# Stochastic simulations reveal few green wave surfing populations among spring migrating herbivorous waterfowl

Xin Wang [1,2], Lei Cao[1,3], Anthony D. Fox[4], Richard Fuller [5], Larry Griffin[6], Carl Mitchell[6], Yunlin Zhao[7], Oun-Kyong Moon[8], David Cabot[9], Zhenggang Xu[7], Nyambayar Batbayar[10], Andrea Kölzsch [11,12,13], Henk P. van der Jeugd[14,15], Jesper Madsen[4], Liding Chen[1,3] & Ran Nathan[2]

Tracking seasonally changing resources is regarded as a widespread proximate mechanism underpinning animal migration. Migrating herbivores, for example, are hypothesized to track seasonal foliage dynamics over large spatial scales. Previous investigations of this green wave hypothesis involved few species and limited geographical extent, and used conventional correlation that cannot disentangle alternative correlated effects. Here, we introduce stochastic simulations to test this hypothesis using 222 individual spring migration episodes of 14 populations of ten species of geese, swans and dabbling ducks throughout Europe, East Asia, and North America. We find that the green wave cannot be considered a ubiquitous driver of herbivorous waterfowl spring migration, as it explains observed migration patterns of only a few grazing populations in specific regions. We suggest that ecological barriers and particularly human disturbance likely constrain the capacity of herbivorous waterfowl to track the green wave in some regions, highlighting key challenges in conserving migratory birds.

[1] State Key Laboratory of Urban and Regional Ecology, Research Center for Eco-Environmental Sciences, Chinese Academy of Sciences, 100085 Beijing, China. [2] Movement Ecology Laboratory, Department of Ecology, Evolution and Behavior, Alexander Silberman Institute of Life Sciences, The Hebrew University of Jerusalem, 91904 Jerusalem, Israel. [3] University of Chinese Academy of Sciences, 100049 Beijing, China. [4] Department of Bioscience, Aarhus University, Kalø, Grenåvej 14, DK-8410 Rønde, Denmark. [5] School of Biological Sciences, University of Queensland, Brisbane, QLD 4072, Australia. [6] The Wildfowl & Wetlands Trust (WWT), Slimbridge, Gloucestershire GL2 7BT, UK. [7] College of Life Science and Technology, Central South University of Forestry and Technology, 410004 Changsha, China. [8] Animal and Plant Quarantine Agency, Gimcheon 39660, Republic of Korea. [9] School of Biological, Earth and Environmental Science, University College Cork, Distillery Fields, North Mall, Cork T23 N73K, Ireland. [10] Wildlife Science and Conservation Center, B-802 Union Building, Sukhbaatar District, Ulaanbaatar 14210, Mongolia. [11] Department of Migration and Immuno-Ecology, Max Planck Institute for Ornithology, Am Obstberg 1, 78315 Radolfzell, Germany. [12] Group of Mathematical Modelling, Institute for Chemistry and Biology of the Marine Environment, Carl von Ossietzky University Oldenburg, Carl-von-Ossietzky-Straße 9-11, 26111 Oldenburg, Germany. [13] Institute for Wetlands and Waterbird Research e.V. (IWWR), Am Steigbügel 3, 27283 Verden(Aller), Germany. [14] Vogeltrekstation—Dutch Centre for Avian Migration and Demography (NIOO-KNAW), Wageningen 6708 PB, The Netherlands. [15] Sovon Dutch Centre for Field Ornithology, PO Box 65216503 GA Nijmegen, The Netherlands. Correspondence and requests for materials should be addressed to L.C. (email: leicao@rcees.ac.cn) or to R.N. (email: ran.nathan@mail.huji.ac.il)

 

Long-distance migration reflects animal responses to large-scale spatial and temporal changes in environmental factors[1]. Tracking the seasonal availability of optimal food resources is generally considered as a widespread phenomenon[2] and the main proximate driver of migration, as exemplified by the green wave hypothesis[3,4] and the closely related forage maturation hypothesis[5]. Both hypothesise that spatial and temporal changes in foliage quality drive the progress of migration of herbivores and predict that the timing of migration links to foliage phenology. Field studies[6–9] support these hypotheses. More extensively, empirical relationships between migration and vegetation indices derived from remote-sensing techniques, using migration data derived from remote telemetry devices[10–12], citizen-science data[13] or weather surveillance radar data[14], as well as experimental approaches[15], also support such hypotheses. These studies used various vegetation greening metrics, such as the normalised difference vegetation index (NDVI)[16,17], green wave index (GWI, the scaled NDVI) and instantaneous rate of green-up (IRG, the acceleration of time-NDVI curve) calculated by fitting annual time-NDVI curves[10,11,18].

The application of migration–vegetation correlational studies to various species has prompted the widespread acceptance of the green wave as the proximate mechanism underpinning herbivore migration patterns[11,12,19]. Nevertheless, two major sources of doubt remain. First, because almost all studies draw conclusions based on correlations, it is impossible to determine whether the green wave is indeed the major determinant of migration patterns, or if such significant correlations arise coincidentally. Although the northward spring migration of northern hemisphere herbivores coincides with food availability (i.e., the green wave), avian spring migration could be associated with other environmental factors, such as day length and air temperature[20], which correlate with latitude. Such multiple associations cannot be disentangled using correlations alone. Testing whether the green wave determines spring migration requires going beyond correlations, to estimate the probability of detecting a match against the corresponding random (null) expectations of directional northward movement irrespective of the progress of the green wave. Here, we derive such null models (sensu Gotelli and Graves[21]) using stochastic simulations (see Methods).

Second, the generality of herbivores tracking the green wave, and evidence to support it, requires confirmation across multiple populations, species and geographical regions[22]. Previous studies focused on single species within restricted geographical ranges[10–12], while comparative analyses across populations, species and regions are lacking. We therefore ask: does the green wave hypothesis represent a general mechanism pertinent to species belonging to various feeding guilds through indirect food web effects[2], or, more simply, to all migratory avian herbivores? More specifically, while the hypothesis might explain grazing herbivore migration well, it might be expected to provide weak or no support for other herbivores or omnivores that tend to graze more facultatively[23]. Alternatively, the hypothesis might not apply robustly even to grazers, as they can occasionally exploit more diverse food items such as non-leaf and even non-plant material[23,24], and migration timing can be affected by various other extrinsic (e.g., weather[25] and competition[26]) and internal factors (e.g., fat deposits[27]).

It is therefore important to examine whether the green wave represents a general mechanism that can explain spatio-temporal migration patterns of herbivorous waterfowl across species and geographical regions. To this end, one needs to test the conventional migration–green wave associations and compare empirical and simulated data, across multiple populations of several species. In this way, one can examine whether the green wave hypothesis explains spring migration of herbivorous waterfowl, or whether spring migration patterns simply reflect stochastic directional movement towards species' breeding grounds irrespective of the progress of the green wave. Specifically, we propose the following three predictions, arranged by decreasing level of support (i.e., from the ubiquitous to the particular) for the green wave hypothesis:

Prediction 1: the green wave is the fundamental driver of spring migration of *all* studied herbivorous waterfowl, leading to significant migration–green wave associations for all populations of all study species; alternatively,

Prediction 2: the green wave only holds for grazers, hence significant associations are expected for all populations of all known conventionally defined grazers, but not for other herbivores and omnivores; or finally,

Prediction 3: the green wave might not explain migration patterns for all grazers, and migration–green wave associations are inconsistent among different populations of grazing species.

To assess the level of support for these predictions, one can first evaluate the robustness of migration–green wave associations using three methods: the Simple Conventional Correlation[12,20], the Correlation method evaluated by Stochastic Migrations[18] and the Metric Selection approach based on Stochastic Migrations (MSSMs) introduced in this study. The Simple Conventional Correlation method[12,28] uses linear models to test for significant correlations between observed and expected arrival dates (the latter is the day with peak green-up rate) at each stopover site. The Correlation method evaluated by Stochastic Migrations compares the observed-expected vs. simulated-expected correlation coefficients to validate the migration–green wave associations identified by Simple Conventional Correlation. The MSSM calculates the value of green wave metrics at the time observed/simulated birds arrive at each stopover site, and compares these values for observed vs. stochastic migrations (Supplementary Table 1). To build the null models of stochastic migration, one should consider the following three simulation types: (1) stochastic timing, (2) stochastic stopover site (when migration tracks are available) and (3) stochastic timing and stopover site modelling (when migration tracks are available) (Supplementary Table 2). If a population supports the green wave hypothesis, the observed migration should be different from all the three null models, reflecting a spatial-temporal migration–green wave association.

Inherently, because the real association between migration and green wave is unknown, one cannot compare the three methods against known associations. Therefore, two general criteria—derived directly from the most basic argument of the green wave hypothesis (reliance on foliage quality)—were employed to evaluate the results obtained by these methods. The first criterion assumes that the tendency of avian migrants of different species to follow the green wave is determined by their reliance on foliage utilisation, based on conventional classification (grazers > other herbivores > omnivores). The second criterion is based on the same assumption, but the reliance on foliage consumption is based on more subtle differentiation in diet composition, found to correlate strongly with some measures of bill morphology, as demonstrated for grazing Anatidae in particular[24].

In this study, we focus on testing the green wave hypothesis rather than testing a range of alternative explanations, aiming to provide an unequivocal test of the ubiquity of the green wave as a main driver of avian herbivore migration. Our analyses show that among 222 individual spring migration episodes of 14 populations of 10 species of geese, swans and dabbling ducks throughout Europe, East Asia and North America, the green wave hypothesis is supported only for grazing species with a particular bill shape that optimises grazing performance, whereas all other observed migrants did not track the green wave better than simulated

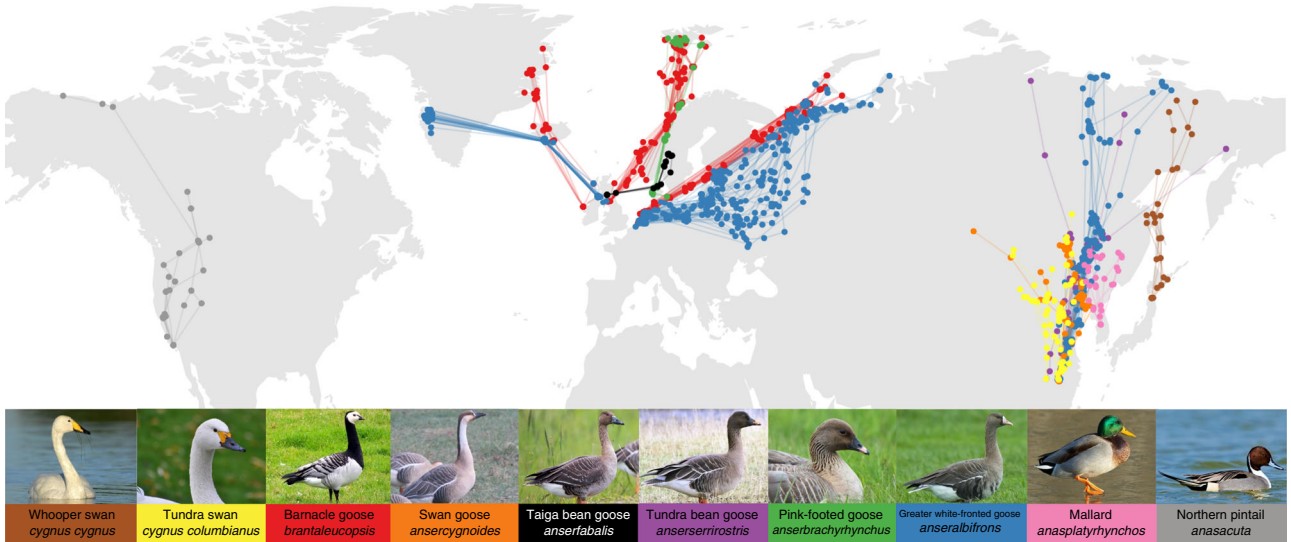

**Fig. 1** Overview of spring migration and stopover site dataset for Anatidae. The dataset includes 222 spring migrations from 193 individuals belonging to 14 populations (five grazers, seven facultative herbivores and two omnivores) of 10 species covering Europe, East Asia and North America, from 1995 to 2016. Points are stopover site locations; point colour corresponds to background colour of species names; consecutive stopover sites during the individual migrations are connected by a line, as an indication of migration route. (Photo credits in order of appearance: J. Frade, M. Langthim, P. Ertl, Y. Muzika, O. Samwald, D. Cooper, B. Keen, S. Harvančík, S. Harvančík and M. Panchal)

stochastic north-migrating ones. Furthermore, this support is inconsistent even among the grazing species and populations, exhibiting marked geographical variation, which leads us to suggest that ecological barriers and human disturbance likely constrain the capacity of herbivorous waterfowl to track the green wave in some regions more than in others.

## Results

**Dataset.** The dataset comprised 222 telemetry-tracked spring migrations from 193 individuals belonging to 14 populations (five grazers, seven facultative herbivores and two omnivores) of 10 species from Europe, East Asia and North America, from 1995 to 2016 (for details, see Supplementary Table 3 and Fig. 1). It contained 125 bird-years (108 birds, some containing multiple-year migrations) of grazing populations, including Greenland, Svalbard and Barents Sea barnacle goose *Branta leucopsis*, Barents Sea and East Asian greater white-fronted goose *Anser albifrons*, 82 bird-years (70 birds) of facultative herbivores including whooper swan *Cygnus cygnus*, tundra swan *Cygnus columbianus*, swan goose *Anser cygnoides*, Scandinavian taiga bean goose *Anser fabalis*, East Asian tundra bean goose *Anser serrirostris*, Svalbard pink-footed goose *Anser brachyrhynchus*, and Greenland greater white-fronted goose *Anser albifrons flavirostris*, and 15 bird-years (15 birds) from omnivores including East Asian mallard *Anas platyrhynchos* and North American northern pintail *Anas acuta* (see Supplementary Fig. 1, for example, stochastic migrations). All populations migrated north during spring in a fairly smooth manner in Europe and Siberia, and more heterogeneously in East Asia (Supplementary Fig. 2).

**Method evaluation.** The Simple Conventional Correlation method yielded weak migration–green wave associations classified as weak surfers for three out of five grazers, for one out of seven facultative herbivores, and for one out of two omnivores. One facultative herbivore showed strong migration–green wave associations hence classified as surfer. The other populations were classified as non-surfers (Table 1, Fig. 2, Supplementary Fig. 2). Surfers, weak surfers and non-surfers were defined as cases of

strong, weak or no significant migration–green wave regression, a terminology used only regarding the Simple Conventional Correlation method (for details, see the section on Simple Conventional Correlation of arrival time tests of migration–green wave of the Methods and Supplementary Table 1). The best models based on Simple Conventional Correlation did not include bill morphology as a predictor (Supplementary Table 4). Given its intrinsic variable multicollinearity problem, and according to the two criteria, the suitability of Simple Conventional Correlation for testing the green wave hypothesis is questionable. However, this method showed how the migration progress was associated with the green wave. The tracked birds migrated faster and arrived earlier than the green wave at stopover sites (Fig. 2, Supplementary Table 5), as all slopes were less than one and the data points mostly clustered within the lower-right part of the panel, below the expected 1:1 relationship. Most populations showed high between-individual variation in the migration–green wave correlation (Supplementary Table 5, Supplementary Fig. 3).

The Correlation method evaluated by Stochastic Migrations suggested that all the surfers and weak surfers but one omnivore identified by Simple Conventional Correlation were indeed green wave surfers. However, this test failed to meet the two evaluation criteria (i.e., neither feeding guild nor bill morphology explain this pattern; Supplementary Table 4). Hence, despite its design to resolve the problem of multicollinearity, the Correlation method evaluated by Stochastic Migrations should be carefully considered in further tests of the green wave hypothesis.

The MSSM method identified three grazing populations (Svalbard and Barents Sea barnacle goose and Barents Sea greater white-fronted goose) as green wave surfers. For these populations, both spatial (i.e., metrics of observed migrations differed from stochastic stopover site migrations) and temporal (i.e., metrics of observed migrations differed from stochastic timing migrations) simulations showed significant deviation from stochastic migration in the hypothesised direction using the metric instantaneous rate of green-up (IRG, Table 1, Figs. 2–4 and Supplementary Figs. 4–6). All other populations, including three populations (whooper swan, pink-footed goose and northern pintail) identified as green wave surfers or weak surfers by the Simple Conventional

**Table 1 Comparison of three different methods for assessing the level of support for the green wave hypothesis**

| Species | Population | Feeding guild | Simple Conventional Correlation[a] | Correlation method evaluated by Stochastic Migrations[a,b] | MSSM[c] |
|---|---|---|---|---|---|
| Barnacle Goose *Branta leucopsis* | Greenland | Grazer | ○ | ○ | ○ |
| Barnacle Goose *Branta leucopsis* | Svalbard | Grazer | ◖ | ● | ● |
| Barnacle Goose *Branta leucopsis* | Barents Sea | Grazer | ◖ | ● | ● |
| Greater White-fronted Goose *Anser albifrons* | Barents Sea | Grazer | ◖ | ● | ● |
| Greater White-fronted Goose *Anser albifrons* | East Asia | Grazer | ○ | ○ | ○ |
| Whooper Swan *Cygnus cygnus* | East Asia | Facultative herbivore | ● | ●[d] | ○ |
| Tundra Swan *Cygnus columbianus* | East Asia | Facultative herbivore | ○ | ○ | ○ |
| Swan Goose *Anser cygnoides* | East Asia | Facultative herbivore | ○ | ○ | ○ |
| Taiga Bean Goose *Anser fabalis* | Scandinavia | Facultative herbivore | ○ | ○ | ○ |
| Tundra Bean Goose *Anser serrirostris* | East Asia | Facultative herbivore | ○ | ○ | ○ |
| Pink-footed Goose *Anser brachyrhynchus* | Svalbard | Facultative herbivore | ◖ | ● | ○ |
| Greater White-fronted Goose *Anser albifrons* | Greenland | Facultative herbivore | ○ | ○ | ○ |
| Mallard *Anas platyrhynchos* | East Asia | Omnivore | ○ | ○ | ○ |
| Northern Pintail *Anas acuta* | North America | Omnivore | ◖ | ○ | ○ |
| Criteria to evaluate the level of support[e] | | | | | |
| Differences among feeding guilds | | | ✗ | ✗ | ✓ |
| Effect of bill morphology | | | ✗ | ✗ | ✓ |

Scientific names of species were shown in italics. The three methods are Simple Conventional Correlation of arrival time, Correlations based on Stochastic Migrations, and Metric Selection approach based on Stochastic Migrations (MSSMs). Level of support (from high to low) is marked as ● for a surfer, ◖ for a weak surfer in Simple Conventional Correlation, and ○ for a non-surfer. ✓/✗ denotes that the results met/failed to meet the evaluation criteria. See Supplementary Table 1 for definitions of surfer, weak surfer and non-surfer
[a]Statistical results are shown in Supplementary Table 5
[b]Results only applicable for green wave surfers or weak surfers identified by Simple Conventional Correlations
[c]Based on results using the instantaneous rate of green-up (IRG) metric; statistical results are shown in Figs. 2–4 and Supplementary Figs. 4–6
[d]Supported by stochastic timing migrations, the only stochastic migration because of the lack of migration tracks for simulation of the other two types of stochastic migrations
[e]Statistical results are shown in Supplementary Table 4

Correlation method showed no significant migration–green wave associations according to the MSSM method. The upper mandible depth-length ratio was found to play significant role in determining migration–green wave associations evaluated by MSSM (ΔAICc = 2.71, Supplementary Table 4). Overall, because the MSSM approach fulfilled both evaluation criteria, whereas both Simple Conventional Correlation and Correlation method evaluated by Stochastic Migrations did not, we focus on MSSM results in testing the green wave hypothesis and our three predictions.

**Testing the green wave hypothesis**. The MSSM approach suggests that green wave surfers selected the peak value of the IRG, which approximately concurs with 50% GWI, rather than the peak values of other metrics. The approach also showed that both day length and air temperature were not significantly associated with the migration of any population (Fig. 2). This method rejects the hypotheses asserting that the green wave could explain spring migration of birds belonging to different feeding guilds (prediction 1) or all avian grazers (prediction 2). However, some grazers supported the green wave hypothesis (prediction 3). We used mixed-effect logistic models with the level of support for migration–green wave as the response variable, the bill morphology and biological family as fixed effects, and geographical range as random effects (for details, see Methods). Geographical region also emerged as a strong predictor of the tendency to follow the green wave (bill morphology explained 69.8% of total variance, and geographical explained 28.2%, leaving only 2.0% of total variance unexplained, Supplementary Table 4). This is in

line with the marked geographical structure in the distribution of green wave surfers. Both of the two grazers in Western Europe— Barents Sea, the barnacle and greater white-fronted geese, were green wave surfers. In Western Europe—Svalbard, the barnacle goose, the only grazer, was a green wave surfer, whereas the pink-footed goose, a facultative herbivore, was not. In Western Europe —Greenland, neither the barnacle nor greater white-fronted geese surfed the green wave. In East Asia, the only grazer, the greater white-fronted goose, was not a green wave surfer. All populations but this one that did not surf the green wave showed the same IRG values by following alternative timing, i.e., the IRG values are nonsignificant between the observed and stochastic timing migrations. This suggests that these non-surfers cannot obtain improved green wave metrics within the overall constrained migration time window. The results also showed that the East Asian grazer and facultative herbivores modelled using stochastic stopover sites apparently followed the green wave better than the observed ones: the GWI values by stochastic migration was higher (Fig. 2 and Supplementary Figs. 4–6). A similar result held for the NVDI metric (excepting the bean goose) but was less clearly seen in the IRG.

**Discussion**
Using a large dataset of spring migrations of geese, swans and dabbling ducks, and a broad set of green wave metrics, we tested the green wave hypothesis using three methods, Simple Conventional Correlation, Correlation method evaluated by Stochastic Migrations and the introduced MSSM. Only the MSSM approach met the general expectations of the green wave

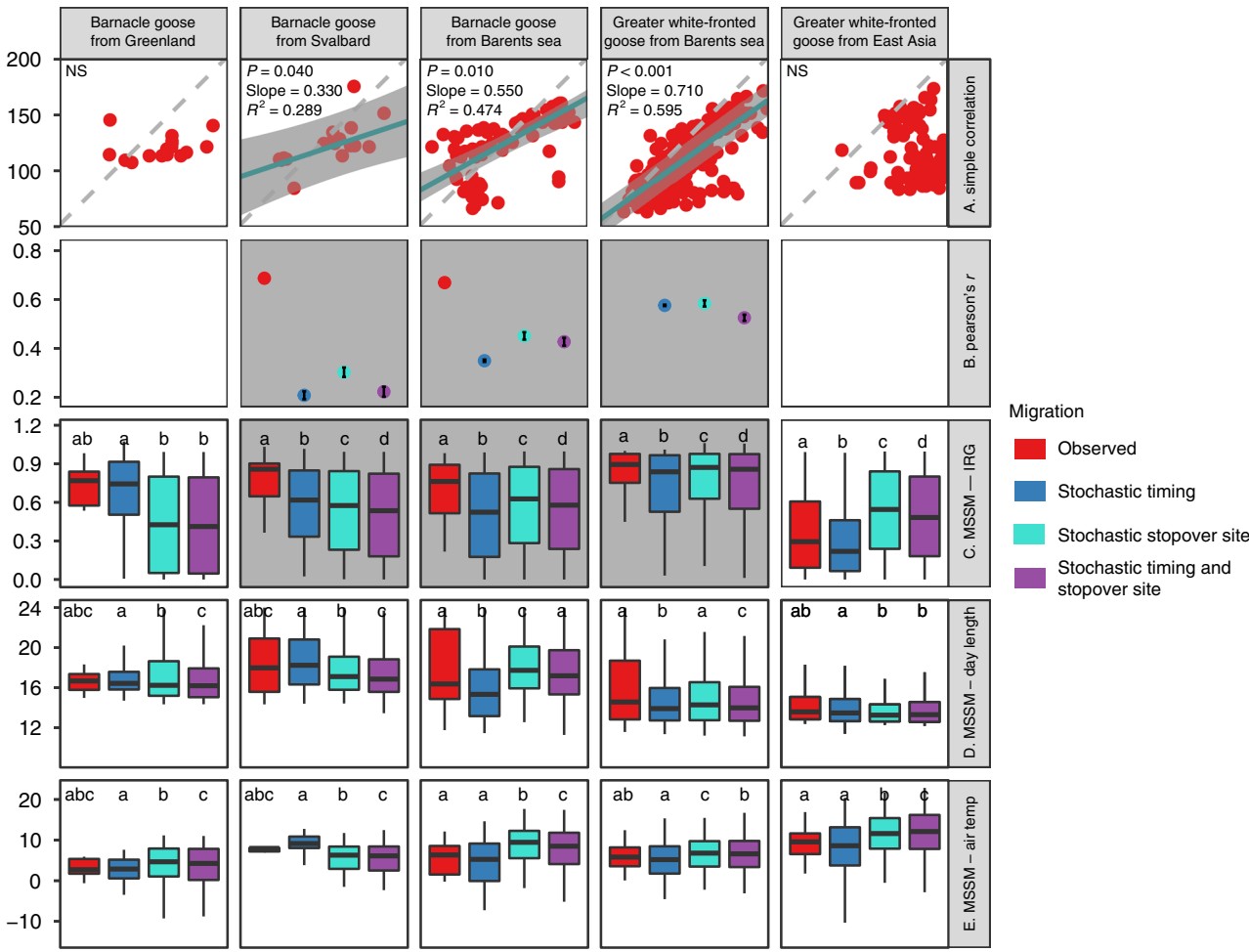

**Fig. 2** Testing the green wave hypothesis for grazers by three different methods. We present the results of the Simple Conventional Correlation (a, upper row), Correlation method evaluated by stochastic migrations (b, second row) and the Metric Selection approach based on Stochastic Migrations (MSSMs) (c–e, three lower rows). Red, blue, turquoise and purple dots/boxes denote observed, stochastic timing, stochastic stopover site and stochastic timing and stopover site migrations, respectively. **a** The x and y axes denote the expected arrival day of the year at stopover sites (the day with peak instantaneous rate of green-up [IRG] value) and the observed arrival day of the year by birds, respectively. The grey pecked lines with slope = 1 and intercept = 0 indicate perfect match of migration and green wave. N.S. denotes insignificant slope; otherwise the p-value and coefficient of the slope, and marginal $R^2$ are provided. Blue lines show the significant positive slope of the green wave in models of green wave surfers, and grey bands are the prediction intervals of the models. **b** Pearson's correlation coefficient r and 95% CI (y axis) of observed and stochastic migrations (x axis). For populations without available migration tracks, only stochastic timing simulations were performed, compared and plotted. Blank panels denote not applicable because this method only applies to green wave surfers or weak surfers identified by Simple Conventional Correlation. **c–e** Three metrics compared for observed versus stochastic migrations: IRG (instantaneous rate of green-up), day length and air temperature. Lower case letters indicate significantly different groups using Kruskal–Wallis test followed by Dunn's test of multiple comparisons. Boxplots show median, first and third quartiles with whiskers reaching to the last data point within 1.5 × interquartile range. For clear presentation, outliers out of 10 and 90% quantiles were excluded from the plots but kept in all analyses. All grey shaded plots in all panels denote significant migration–green wave associations. Source data are provided as a Source Data file

hypothesis, and revealed that only a few of the grazer populations followed the green wave during spring migration. Hence, although the green wave hypothesis can explain migration patterns of some avian grazers, it does not represent the predominant proximate mechanism determining spring migration progress in the studied herbivorous waterfowl populations at large geographical scales. We therefore conclude that within the taxonomical and geographical range covered in this study, tracking the green wave is neither a global phenomenon nor a ubiquitous driver of spring migration of herbivorous waterfowl.

Our study suggests that the variation in the associations between migration patterns and the green wave can, to some degree, be related to variation in bill shape of herbivorous waterfowl. Bill shape correlates with waterfowl diet composition[24],

specifically, with the level of dependence on grazing green leaves/shoots. Herbivorous bird species with higher bill depth-length ratios are more effective grazers, but less able to exploit other sources of plant food[24]. Therefore, species with large bill depth-length ratio are more likely than other species to migrate with the green wave during spring, to match closely the shift in the timing and location of suitable grazing sites along their routes. Our stochastic simulations supported the green wave hypothesis chiefly for the most exclusive grazer, the barnacle goose, with the highest depth-length ratio and the largest proportion (2/3 populations) of significant migration–green wave associations. The only other species for which stochastic simulations supported this hypothesis was the greater white-fronted goose (with the second largest bill depth-length ratio in our sample), whereas the East Asian

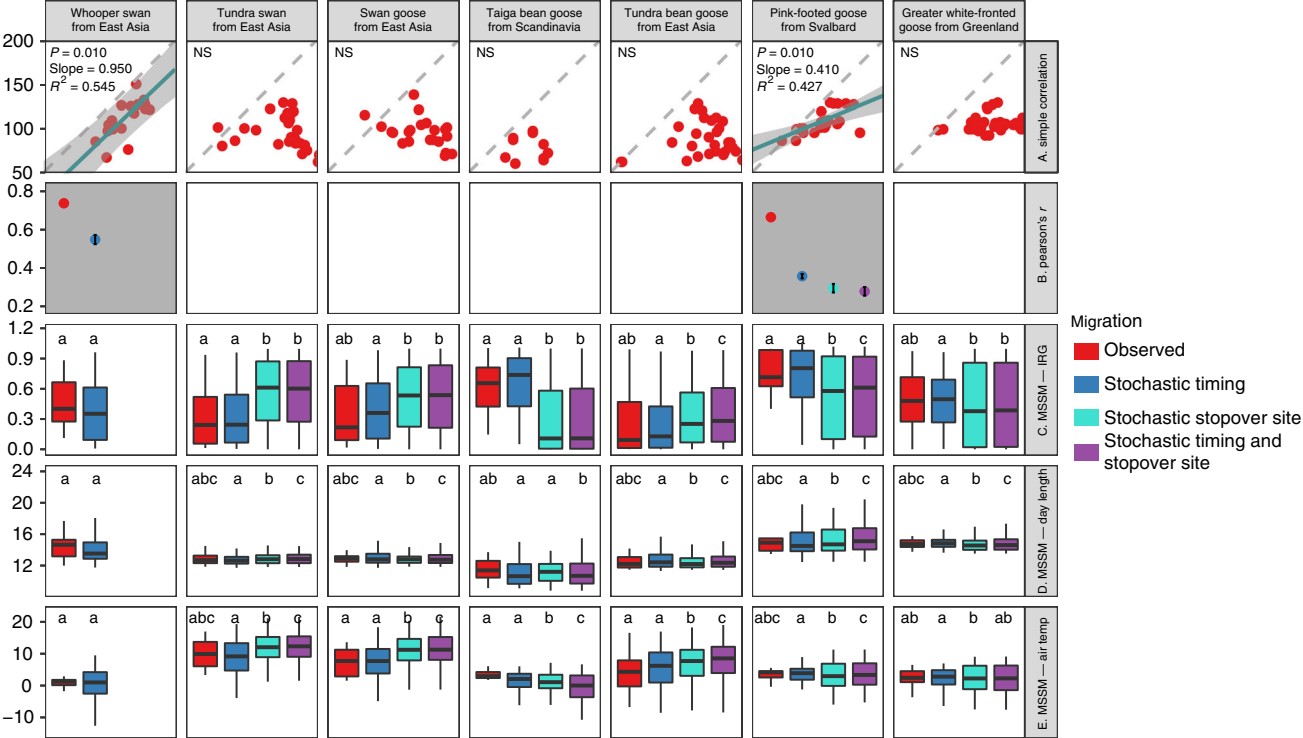

**Fig. 3** Testing the green wave hypothesis for facultative herbivores by three different methods. See Fig. 2 for definitions and details of panels, symbols, colours, acronyms and boxplots. For the population without available migration tracks (Whooper Swan from East Asia), only stochastic timing simulations were performed, compared and plotted. Source data are provided as a Source Data file

population of that species (and all other populations of all other species) with smaller ratio showed no migration–green wave associations.

The exception of the East Asian greater white-fronted goose from this otherwise general bill shape effect was notable. Furthermore, our analyses showed no support for the green wave hypothesis also for all other five populations of geese, swans and ducks in this region. We suggest that human disturbance, which plays an important role in determining the progress of bird migration[29,30], could explain the geographical deviations from the green wave hypothesis, especially in East Asia. In this region, hunting pressure, land use change, poisoning and other human disturbance are intensive[31,32]. Five wintering goose species were almost entirely confined to natural wetlands in the Yangtze River Floodplain[33], whereas geese elsewhere in the world commonly use energetically profitable farmland habitats[34,35]. This could explain our finding that for all East Asian species included in our analyses, simulated spatially stochastic migrants obtained higher green wave metric values than the observed tracks (Figs. 2–4 and Supplementary Figs. 4–6), indicating remarkably poor (worse than random) selection of stopover sites by the birds in relation to the green wave. Furthermore, tracked birds arrived consistently earlier than the green wave at stopover sites (Fig. 2, Supplementary Fig. 2, Supplementary Table 5). To explain these results, we propose the following scenario. Birds at initial (early spring) stopover sites departing northwards skip subsequent stopover sites along their migration route with high forage quality (i.e., green wave phenomenon) due to excessive human disturbance there (e.g., habitat loss or physical human activity), forcing the birds to undertake longer jumps to the next stopover sites. Consequently, the birds arrive to their next stopover sites prior to the date of optimum availability of local food resources at that latitude (i.e., ahead of the green wave) and stage longer in these sites. This proposed explanation is supported by the lack of

stopover sites between the Yangtze wintering ground and northeast China, a huge area with potentially good green wave conditions rendered apparently inaccessible to migrating birds (Supplementary Fig. 2). This is also corroborated by the finding that waterbirds in this flyway progress northward in jumps double the length of those in the western Palearctic flyway[36]. Furthermore, early arrival to stopover sites ahead of the green wave would be expected to lengthen stopover duration, potentially explaining the ~25% longer stopover duration found for greater white-fronted geese in East Asia compared with the Barents Sea (mean ± SD  12.17 ± 11.37  vs.  9.79 ± 8.57 days, respectively; t-test, $t = -2.0403$, df = 183.31, $p = 0.043$). Hence, habitat loss, excessive human disturbance and hunting in East Asia and some other parts of the world might challenge the ability of migrating waterbirds to surf the green wave.

In contrast to the greater white-fronted goose and five facultative herbivores of East Asia, two facultative herbivores outside this region—the bean goose from Scandinavia and pink-footed goose from Svalbard—would obtain lower (rather than higher) IRG values by spatially stochastic migrations. This suggests selection for stopover sites of high forage quality during migration, as leaf material comprises a considerable part of the diet of these two populations during migration[37,38]. However, birds of these two populations also feed on other food types such as grains during stopover[37,38], so that their migrations were not constrained by the green wave. This explains the lack of difference in the green wave metrics between the observed and stochastic timing migrations, implying that observed birds did not tightly coordinate the timing of their migration with the green wave. Alternatively, the weak green wave surfing in these two populations might be attributed to other factors, such as ecological barriers and environmental predictability (discussed further below). However, such explanations seem unlikely because the Svalbard barnacle goose did surf the green wave within the same

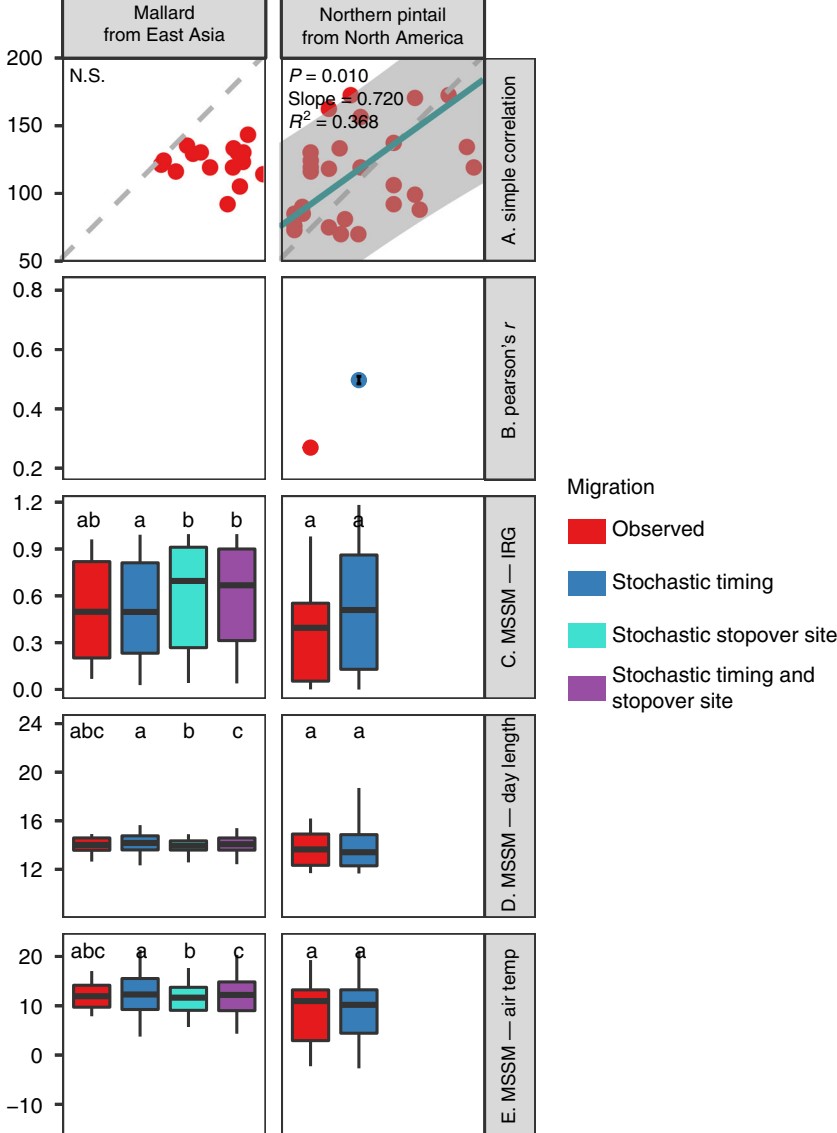

**Fig. 4** Testing the green wave hypothesis for omnivores by three different methods. See Fig. 2 for definitions and details of panels, symbols, colours, acronyms and boxplots. For the population without available migration tracks (Northern Pintail from North America), only stochastic timing simulations were performed, compared and plotted. Source data are provided as a Source Data file

geographical range, obtaining a significantly higher IRG on observed migration compared with stochastic timing (and site, and timing and site) migration.

Migratory herbivorous waterfowl might not follow the green wave for reasons other than human disturbance. First, birds may be limited in their ability to predict the progress of the green wave, especially when and where the next stopover lies beyond a large ecological barrier (e.g., open sea, desert, mountain ranges and ice sheets)[39]. This seems the case especially for north European geese migrating to breed in Greenland, as both the Greenland populations of barnacle and greater white-fronted geese, two species with high bill depth-length ratio, did not surf the green wave in the Western European–Greenland flyways, presumably due to the extended migration over stretches of ocean imposed on these populations. Second, even if birds are capable of perfectly predicting the green wave, unpredictable adverse weather conditions[40–42] can alter migration progress and induce a mismatch with the green wave; migrating birds, for example, might delay departure from a stopover site due to storms or wait for sufficient tailwinds to assist with crossing migratory

barriers[25]. Such effects, however, are more likely to delay than to advance arrival to stopover sites, hence contradict the finding that the likelihood of early arrival increases with distance to the next stopover site. Third, the advancement in peak food availability caused by climate change could exacerbate the mismatch between migration timing and phenology, because of the intrinsic time constraint on spring migrations[43]. Fourth, birds might progressively overtake the green wave to arrive well before the peak in food availability to produce a clutch of eggs that hatch to coincide with the peak in gosling food quality[43,44]. These factors in turn introduce further variation in migration timing, and potentially a larger mismatch between the green wave peak and migration timing (Supplementary Figs. 2 and 3). Such individual mismatches can be quantified using approaches such as the space-time-time matrix[10]. However, such between-individual variation seemed to be negated within populations, and some population-level migrations still showed predictable associations with the green wave.

Comparing our results to those of previous studies requires a methodological comparison as the first step. In addition to the

commonly applied Simple Conventional Correlation method and the more rarely used Correlation method evaluated by Stochastic Migrations method, we introduced the MSSMs. Overall, the MSSM approach was found to provide a much more conservative test to the green wave hypothesis compared with the other methods. Furthermore, the two other approaches tended to overestimate migration–green wave associations, as three populations—the whooper swan, the pink-footed goose and the northern pintail—were identified as green wave surfers or weak surfers by at least one of these methods, but showed no significant association according to the MSSM method. To our knowledge, no studies on avian migration have yet used stochastic spatio-temporal simulations to test the green wave hypothesis. Aikens et al.[18] generated stochastic (null) migration models of ungulates using a coarse method (as they admitted), and tested their results using the Correlation method evaluated by Stochastic Migrations, which we found less suitable compared with MSSM in our study (Table 1). Bridge et al.[45] constructed random migrations, without statistically comparing stochastic vs. observed metrics. Considering its reliable performance and ability to generate null expectations while controlling for confounding effects of potential driving factors common in animal movement studies, we advocate the use of the MSSM approach for investigating questions relating to environmental drivers of migration and other movement types for animals.

Another important methodological comparison is among NDVI-derived metrics of vegetation phenology. Although each reflects certain features of vegetation, such as nutrition or quantity, no single index alone can capture the fine details of migrants' food availability in relation to requirements, which renders the a-priory use of any single index to test migration–green wave associations questionable. We therefore used a series of metrics, which highlighted the more reliable performance of the IRG compared with other indices. Using multiple green wave metrics in combination to test the green wave hypothesis may better represent vegetation features and animal demands on the vegetation, hence should be examined in future studies. In addition to the selection of different metrics, differences in spatial or temporal resolution and scale used to estimate these metrics might affect the study outcomes. For example, Kelly et al.[14] attributed the lack of support for the green wave hypothesis to the lack of seasonal south-to-north greening derived from remotely sensed indices estimated in spatial resolution units of 80 km radius[14]. In our study, the typical spatial extent of a stopover site was 5–10 km radius across species and populations. This implies that an 80 km radius typically encompasses an area roughly 63–255 times larger than the relevant area for estimating stopover site characteristics of migratory ducks, geese and swans. Nevertheless, possible resolution and scale effects on the robustness of migration–green wave associations await further investigation.

Methodological differences also occur in other dimensions of research practice and design. Both field manipulations aimed at testing the green wave hypothesis[6,15], and meticulous empirical attempts to measure vegetation quality at each observed stopover site[6] (rather than derive estimates from remote sensing), inherently remain limited practically in time and space. Furthermore, as discussed above, the spatio-temporal resolution of citizen-science data[13] or weather surveillance radar data[14] may not be sufficient to test the green wave hypothesis. The alternative approach, taken in this study—to couple data from advanced wildlife tracking technologies and remote sensing—has strong merits[46,47] but important limitations as well[48], including difficulties in trapping wild animals, adverse effects of trapping and tagging on animal behaviour and fitness, restricted sample sizes, bias to relatively large-bodied animals, and high costs. For these

reasons, it is challenging to obtain sufficiently large sample sizes to achieve good taxonomical and geographical representativeness in wildlife tracking research. We have concentrated on long-distance migratory (mostly arctic nesting) herbivorous waterfowl and although the dataset of this study is by far larger than previous green-wave studies, the 10 study species are unlikely to adequately represent all 169 global species of Anatidae[49], nor can one study population (northern pintail) represent all North America migratory waterfowl. Cost-effective use of wildlife tracking and other approaches should consider a broader coverage of the major migration flyways, though the paucity of datasets from North America might stem from restricted data availability rather than genuine data gaps. Furthermore, advances in wildlife bio-logging technologies offer complementary auxiliary data from accelerometers and other sensors might enable examining more direct links between the internal state (e.g., energy balance, behavioural context) and the external environment (e.g., meteorological conditions, land use) that individual migrating birds experience en route. In addition to bird movement and behaviour obtained by biotelemetry devices, an assessment of the human-induced changes in habitat use and bird distribution, site/food availability using multiple data source, is critical to evaluating the impact of human disturbance on migratory birds, most notably in East Asia.

Overall, the results of our study, along with the various methodological issues we discussed, cast doubts on the ubiquity of the green wave as a main driver of (spring) migration of herbivorous birds, despite previous support from studies of single species[12,28]. Further broad multispecies comparisons using the MSSM or similar methods are needed to assess the generality of this conclusion in a broader taxonomical and geographical context. These will set the stage to test rigorously not only the green wave hypothesis, but alternative explanations (such as human disturbance, as proposed in this study) as well, to further elucidate the mechanisms driving bird migration and to better conserve migratory birds.

## Methods

**Migration data**. We collected Anatidae migration data from two sources: published and our own movement information first presented in this study. We undertook the literature search on the Web of Science on 18 March 2016 with the terms: (GPS OR Argos OR PTT OR CTT OR (satellite* AND (track* OR transmitter* OR telemetr*))) AND (screamer* OR "magpie goose" OR "magpie geese" OR teal* OR shelduck* OR sheldgoose OR sheldgeese OR anas OR waterfowl OR wildfowl OR anseriformes OR waterbird* OR duck* OR goose OR geese OR swan*). We excluded migration data derived from geolocators or observations from banded individuals because of large spatial or temporal errors and biases involved in using these approaches.

We used movement tracks derived from our tracking studies, involving birds caught in several places in East Asia from 2014 to 2016, and in Europe from 2008 to 2016, and tracked using different types of tracking devices (see Supplementary Data 1 for details of our tracked birds). We also searched and accessed published Anatidae movement tracks on the Movebank Data Repository (https://www.datarepository.movebank.org/).

We excluded maritime populations from search results because their sub- and inter-tidal marine vegetation habitats are not always associated with land, so no specific vegetation metrics were relevant for these species. Because the simulated stochastic migrations required stopover duration for simulation, we excluded literature records, which only reported birds' arrival date. Paired adults and parent geese with their offspring frequently migrate and move together. To avoid pseudo-replication of stopovers caused by such associations, we removed all replicates except for one randomly selected movement tracks that were otherwise temporally and spatially identical.

We classified feeding guilds using literature and observations of the corresponding populations (Supplementary Table 3). We classified populations according to the geographical range during migration. For biological and statistical representativeness, we excluded manipulated birds, such as translocated birds[50], birds that performed incomplete migrations of <1000 km, or populations containing fewer than five individuals or fewer than 10 stopover sites to avoid type II error induced by small sample size. We also excluded tracks that contained gaps longer than 10 days within migration legs, which can cause a biased estimate of arrival date and problems in modelling the migration process using

continuous-time correlated random walk (see below). We excluded migrations in 1994 because 16 out of 52 weekly NDVI images in this year were missing (due to unavailable satellite data), and therefore we could not generate reliable green wave values based on the double-logistic model (see below).

**Stopover/migration information**. We extracted stopover information from the literature that recorded both timing (arrival and departure date) and stopover site coordinates during a tracked individual's spring migration. Where stopover site locations were presented using maps rather than providing coordinates, we determined coordinates of stopover sites from maps in combination with the relevant literature descriptions.

From individual movement tracks, we identified each stopover site and recorded arrival and departure dates. We followed the methods of van Wijk et al.[20], Shariati-Najafabadi et al.[12] and Kölzsch et al.[51] to determine stopover sites, where birds stayed longer than 48 h. In this way, we excluded those stopover sites where birds probably only rested briefly or drank[52]. According to empirical migration observations[22,53], we only included stopover locations after the 60th day of the year, to eliminate within-winter movements.

In addition, full migration tracks from published sources such as Movebank Data Repository and tracking data held by the authors of this study were used for the use of stochastic stopover site modelling and stochastic timing and stopover site modelling (see below). We could not extract full migration tracks from the literature, and therefore the simulations of stochastic stopover site modelling and stochastic timing and stopover site modelling for the whooper swan from East Asia and the northern pintail from North America were not applicable in our analyses.

**Remote-sensing data and green wave metrics**. We used the Advanced Very High Resolution Radiometer Vegetation Health smoothed NDVI Product (AVHRR-VHP)[54] for extraction of vegetation data and analyses. The critical advantage of this product is its larger temporal coverage (1989 onward) compared with MODIS-based products (2000 onward), permitting migration–green wave association analyses using migration data before 2000 extracted from the literature. Moreover, the AVHRR-VHP provides a higher temporal resolution (7 day) than MODIS-based products (8 day or 16 day), providing finer grained parameter estimates for annual NDVI models (see below for more details), which are critical for deriving green wave metrics. We included all stopover sites in the latitudinal range of AVHRR-VHP (55.152°S–75.024°N). The coarser spatial resolution unit of AVHRR-VHP (4 km, in comparison with 250 m to 8 km for MODIS) is unlikely to bias our analyses, because vegetation data at stopover sites were extracted from within a buffer (radius) of 5 to 30 km, conforming with the movement range at stopover sites (Supplementary Table 3), compared with 15 km[12] or 50 km[28] in previous studies. A sensitivity analysis showed that changing the buffer size from 5 to 30 km did not alter our conclusions (Supplementary Table 6). In this article, we reported results based on the 5-km buffer size, in accordance with the movement range within stopovers of the birds (Supplementary Table 3).

We extracted the NDVI values from pixels occupying a 5-km radius buffer around each stopover site identified by a point location from the birds' positional data, accounting for any non-vegetated area within the buffer[55]. We excluded pixels within the buffer classified as forest, woodland, urban and built or bare ground by the AVHRR land cover product[56]. Overall, we obtained $9.8 \pm 1.5$ (SD) pixels per stopover site. Within each year, we fitted a time series model to the scaled NDVI values for each pixel within the buffer using a double-logistic model[57], the method that performed best in a comparison of NDVI filtering approaches[58]

$$\text{NDVI}(t) = \alpha + (\beta - \alpha) \cdot \left( \frac{1}{1 + e^{-\gamma \cdot (t - \delta)}} + \frac{1}{1 + e^{\varepsilon \cdot (t - \theta)}} - 1 \right) \quad (1)$$

where $\alpha$ and $\beta$ are minimum and maximum NDVI values; $\gamma$ and $\varepsilon$ are the rates of increase/decrease of the curve at the inflection points; $\delta$ and $\theta$ are time of maximum/minimum green-up rate (Supplementary Fig. 7). We calculated the GWI at time $t$ when a bird arrived at a stopover site using fitted values following the method by White et al.[59] and Beck et al.[60]:

$$\text{GWI}(t) = \frac{\text{NDVI}(t) - \alpha}{\beta - \alpha} \times 100\% \quad (2)$$

We calculated the predicted NDVI and IRG (the first derivative of NDVI time series rescaled from 0 to 1), based on Eq. (1). We also excluded pixels with anomalous spring phenology where the maximum green-up rate occurred (1) before the 50th day of the year, or (2) after the 240th day of the year, or (3) later than the time when the minimum green-up rate occurs, which often signifies essentially non-vegetated areas. We used the method of Teets[61] to calculate the day length at specific locations and times. The air temperature data were obtained from the global land data assimilation system, on a fixed grid of 0.25° × 0.25° and at a 3-h temporal resolution[62–64]. We calculated daily mean temperature from pixels occupying a 5-km radius buffer around each stopover site, on the arrival day of the birds.

**Simple Conventional Correlation**. Using all the individual stopover data, we performed mixed-effect linear models using maximum likelihood estimates for

each population to conduct Simple Conventional Correlation tests of migration and the green wave with the following structure

$$\text{Day}_{\text{obs}} \sim \text{Day}_{\text{pred}} + \left( 1 + \text{Day}_{\text{pred}} | \text{year/bird} \right) \quad (3)$$

where $\text{Day}_{\text{obs}}$ is the observed arrival day at a stopover site and $\text{Day}_{\text{pred}}$ is the predicted arrival day (the day with peak green-up rate, i.e., 50% GWI) at that site based on Eq. (1), $(1 + \text{Day}_{\text{pred}} | \text{year/bird})$ is the random effect of individual nested in year on both intercept and slope. If all individuals within a population contained only single-year stopover data, we conducted Simple Conventional Correlation tests using the following structure instead

$$\text{Day}_{\text{obs}} \sim \text{Day}_{\text{pred}} + \left( 1 + \text{Day}_{\text{pred}} | \text{bird} \right) \quad (4)$$

A population is designated as green wave surfer when both of the two following conditions are met: (1) significant ($p < 0.05$) positive slope with $1 \geq$ lower 95% CI $> 0$ and upper $\geq 1$ and (2) nonsignificant ($p > 0.05$) intercept, and designated as weak surfer when either of the following two conditions is met: (1) significant slope with lower 95% CI $> 0$ and upper $< 1$, or lower $> 1$ and any intercept, or (2) intercept significantly different from zero ($p < 0.05$) and any significant positive slope[12,18,20,28].

**Correlation method evaluated by Stochastic Migrations**. Correlation method evaluated by Stochastic Migrations compares Pearson correlation coefficients of green wave surfers/weak surfers identified by Simple Conventional Correlation with stochastic migrations. We used (1) stochastic timing modelling, (2) stochastic stopover site modelling and (3) stochastic timing and stopover site modelling to generate 1000 simulated migrations per individual bird to build null models (*sensu* Gotelli and Graves[21]) of the migration–green wave correlation and metric selection. The three null models were used as references for the observed migrations to examine the migration–green wave association for simulated migrations based on random timing, random sites and their combination, respectively (Supplementary Table 2). These null models were used to test whether a metric or an indicator links with migration temporally, spatially, or both, by determining the probability of obtaining the observed parameter values from the expected distributions based on the three stochastic migrations. Because migrations driven by the green wave should show spatial selectivity for sites with a green wave and temporal selectivity for peak time of green-up, observed migrations should differ from all of the three stochastic migrations in terms of the appropriate test metric. Hence, differences from some (but not all) stochastic migrations should not be considered as weak support for the green wave hypothesis.

Under the simulation scheme of stochastic timing modelling, the number and locations of stopover sites and start day of migration for each individual were kept the same as in the individual's observed migration. The stopover duration at each stopover site was randomly drawn with replacement from the stopover duration pool of the population based on the observed stopover durations of the birds. Because of the considerable variation in stopover duration (Supplementary Table 3), the temporal component of migration was well shuffled in the simulations while adhering to the empirical range. The model repeats this process 1000 times to generate 1000 simulations. We rejected random migrations that generated arrival dates that were later than the 210th day of the year.

Under the simulation scheme of stochastic stopover site modelling, the duration of the non-stopover period for migration and the number and duration of stopovers for each individual were kept the same as in the individual's observed migration, but stopover site locations and migration tracks were stochastically generated. Although stochastic movement track simulations have recently been used in several movement ecology studies[65,66], they have never, to our knowledge, been applied to long-distance migration. Two challenges in using such simulations are (1) the contrasting movement pattern of migratory flights (long distance, short time, high speed) and stopover periods (short distance, long time, low speed), which renders successful simulation very difficult; (2) the often irregularly sampled and highly auto-correlated data, which make many movement models inapplicable[67,68].

To address these issues, we performed a four-step process to generate stochastic migration model outputs (Supplementary Fig. 8). First, for each individual's migration in each year, we clipped migration tracks from the last point on the wintering ground to the first point on the breeding grounds or the last stopover site. We removed movement tracks of stopover periods from the complete migration tracks, and fused adjacent segments by directly linking endpoints and set the time interval the same as the interval between the two endpoints from the first endpoint to its subsequent observed point. In this way, we constructed stopover-free migration tracks consisting of flights and short-time stopovers, which were more uniform in terms of movement pattern and easier to model and realistically simulate. For the sake of modelling validity, we excluded fused tracks consisting of fewer than 15 locations.

Second, to preserve individual movement characteristics in random migrations, we applied a continuous-time correlated random walk model (CTCRW) for each individual on the fused migration tracks to estimate parameters for velocity and drift distributions[69]. This approach allows for unevenly sampled tracking data and can handle auto-correlated movement data. By including drift—a less variable movement type (the overall directional migration movement tendencies of larger

spatio-temporal scales, i.e., from wintering to breeding ground), and velocity—a more variable movement type nested within drift (the variable movement patterns at finer spatio-temporal scales, such as the daily flight and roosting movements). For each single fused migration track, we fitted a CTCRW with the starting location of the track, mean migration speed (total displacement between the first and last points of the migration tracks divided by the time accounted for by the fused migration track) and zero as the initial location, drift and velocity, respectively. To obtain biologically meaningful models, we constrained model parameters to only allow decreasing autocorrelation of velocity and drift with increasing time lag (the time interval between any two considered locations), and less variability for drift than velocity, dropping models that were unsuccessful in estimating parameter confidence intervals (see Supplementary Data 2 for model parameters).

Third, we generated 1000 stochastic migrations for each individual in each year based on the model parameters derived in the second step. We initialised the location, drift and velocity with the observed starting location, mean migration speed (as described in the second step) and the difference between observed speed of the first step and the mean migration speed. In the subsequent step, drift and velocity were drawn from distributions determined by the empirical model parameters and the current drift/velocity states (see Johnson et al.[69] for more mathematical details of the model). We ended a simulation when the simulated track lasted the same time as the observed one and kept the simulation if the great-circle distance between the ends of stochastic and observed migration tracks was not >200 km. To make the simulations biologically meaningful, we constrained the maximum overall speed (the combination of drift and velocity) to 120 km per hour[22]. For the sake of speed of simulation and computation, we set the step interval to 2 h. To ensure the simulated migration tracks did not include unlikely migration routes, we estimated the regular migration range of each population based on literature sources[70], eBird database[71] and expert knowledge (Supplementary Fig. 9). We excluded simulated migration tracks if >5% of their locations were outside of the regular migration range.

Finally, we randomly located stopover sites along each stochastic track with the same number of stopover and same durations as the observed tracks. We identified all available points for a stopover site as those which neither fell in the sea within the 5-km buffer zone based on the global shoreline database 2.3.6 at crude resolution[72] nor were classified as forest, woodland, urban and built or bare ground by the AVHRR land cover product[56]. We then randomly chose the locations for stopover sites and designated the corresponding duration. In accordance with the identification of stopover sites, the distance between any stopover sites/wintering/breeding site was not <30 km. The flight distance of any migration leg was not allowed to exceed the maximum flight distance of observed migration legs of the corresponding species.

Under the simulation scheme of stochastic timing and stopover site modelling, the total non-stopover period during the migration and the number of stopover sites for each individual were kept the same as the individual's observed migration, but stopover site locations, migration tracks, migration initiation starting dates and stopover durations were stochastically generated. This simulation scheme was the same as the four-step procedure of stochastic stopover site modelling in the initial three steps but differs in the fourth step where stochastic timing and stopover site modelling performed a stochastic process similar to stochastic timing modelling to generate stochastic stopover timing; that is, for each set of simulated tracks, we randomly located stopover sites along the tracks and then determined migration starting dates and stopover durations from the observed migration starting dates and stopover duration pools of that population, respectively. We only accepted migrations that generated arrival at breeding sites before the 210th day of the year (see Supplementary Table 2 for comparison of the three null models).

We calculated and compared the Pearson's correlation coefficient of the observed arrival day vs. predicted arrival day for the observed migration and three types of stochastic migrations[18]. If (1) the observed migration was classified as a green wave surfer or weak surfer, by Simple Conventional Correlation, and (2) the Pearson's correlation coefficient of the observed migration was positive and significantly higher than those of all types of stochastic migrations, i.e., the observed migration surfed the green wave better than random, then the population was considered as green wave surfer.

**MSSM**. MSSM compares green wave metrics between observed and stochastic migrations generated by (1) stochastic timing modelling, (2) stochastic stopover site modelling and (3) stochastic timing and stopover site modelling, as described above. We calculated all GWIs (fitted NDVI, GWI and IRG), day length and air temperature, for simulated stopover in each simulation scheme and compared this with the observed migrations using Dunn's test of multiple comparisons with $p$-values using Benjamini–Hochberg's adjustment[73]. Because the non-normality of residuals, we could not use the mixed-effect linear model in this case, which can account for the multiple measurements per (observed and stochastic) migration track and year. For populations containing only observed and stochastic timing migrations, we performed Wilcoxon tests to identify differences in the two groups. A population was designated as surfer if the metric of observed migration was significantly higher than all stochastic ones, as suggested by the green wave hypothesis and other studies[12,45].

**Variation in the migration–green wave associations**. We divided the migrations of birds into five geographical regions, based on their flyway and aggregation of stopover sites (Supplementary Fig. 10). Among several bill shape parameters that correlate with dietary composition[24], mechanical advantage, calculated as the ratio of the in-lever (the distance from the bill axis of rotation to the upper mandible-jugal joint) to the out-lever (the distance from the bill axis of rotation to the tip of the upper mandible), correlates significantly with diet composition of individual species. Species with bills with a higher mechanical advantage have a higher proportion of leaves in the diet, and thus are more likely to be associated with the green wave during spring migration. Due to the lack of skeletal specimens for the measures for mechanical advantage calculation, we used the ratio of upper mandible depth-length ratio, which serves as a good approximation for mechanical advantage (adjusted $R^2 = 0.89$, $N = 5$, linear modelling based on the study genus with available mechanical advantage data reconstructed from Olsen[24]). To obtain the upper mandible depth-length ratio, we obtained five lateral photos for each species and measured the bill depth-length ratio (Supplementary Table 7). We used the mixed-effect models, with the existence of migration–green wave associations as the response variable, the approximate mechanical advantage and biological family as fixed effects, and geographical range as a random effect. The three categories of the response variable (non-surfer, weak surfer and surfer) made a direct implementation of the mixed-effect logistic model difficult. We therefore applied three modelling schemes to represent models robust against type II error, with balanced robustness, and robust against type I error respectively: (1) mixed-effect logistic model, considering only surfer as 1 and others as 0; (2) mixed linear model, considering surfer as 1, weak surfer as 0.5, and non-surfer as 0; (3) mixed-effect logistic model, considering surfer and weak surfer as 1 and non-surfer as 0. We included biological family as a fixed factor in the model to account for the cross-family morphological difference, which may introduce systematic bias to the approximated mechanical advantage. We approximated variance partitioning of fixed and random effects using the methods of Nakagawa and Schielzeth[74] and Johnson[75]. We did not use phylogenetic logistic models because our sample size (10 species) was too small to detect phylogenetic signal with sufficient statistical power[76], or to induce type I error[77], and because of the constant association between diet and bill shape irrespective of the use of non-phylogenetic or phylogenetic models[24].

## Data availability
The source data underlying Figs. 2–4 and Supplementary Figs. 4–6 are provided as a Source Data file in figshare (https://figshare.com/s/a65afee29225c6db51cd). The dataset generated and/or analysed during the current study are included in Supplementary Data and Source Data files. A reporting summary for this article is available as a Supplementary Information file.

## Code availability
We conducted all analyses in R 3.3.2[78]. We used the greenbrown package for double-logistic fitting[79], lme4 package for generalised mixed modelling[80] and crawl for continuous-time correlated random walk modelling[69].

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

## Acknowledgements

We thank Lei Fang, Qin Zhu, Hui Yu, Yuzhan Yang and Zhujun Wang for fieldwork; Xianghuang Li, Hongbin Li and Shujuan Fan for collecting remote-sensing datasets; Stuart Pimm, Devin Johnson and Aaron Olsen for constructive comments. We appreciate assistance of the staff of the Poyang Lake National Nature Reserve, Nanjishan National Nature Reserve, East Dongting Lake National Nature Reserve, Shengjin Lake National Nature Reserve, U.S. Geological Survey Western Ecological Research Center, Wildlife Science and Conservation Center of Mongolia and Korea Institute of Environmental Ecology, during the field study. We thank Alyn Walsh, John Skilling, Ed Burrell, Mitch Weegman, Angus Maciver, Arthur Thirlwell and Maurice Cassidy for help with the catching and tagging of the geese in Scotland and Ireland. We thank K.-M. Exo and K. Oosterbeek for supporting obtaining the tracking data of barnacle geese (three populations). We thank David Cooper, Peter Ertl, José Frade, Stanislav Harvančík, Benjamin Keen, Marcel Langthim, Yann Muzika and Manish Panchal for kindly providing images. Bird capture and logger deployment in this study in the UK was carried out under license from the BTO, and elsewhere in accordance with the guidance and permission (no. rcees-ddll-001) of Research for Eco-Environmental Sciences, Chinese Academy of Sciences. The study was supported by the National Key Research and Development Program of China (grant no. 2016YFC0500406), the National Natural Science Foundation of China (grants no. 31661143027, 31670424, 31500315), China Biodiversity Observation Networks (Sino BON), Scottish Natural Heritage, FlySafe (http://iap.esa.int/flysafe)—a project of the European Space Agency Integrated Applications Promotion (IAP) programme (http://iap.esa.int/), the Israel Science Foundation (grant no. 2525/16), the Minerva Center for Movement Ecology, the Adelina and Massimo Della Pergola Chair of Life Sciences and China Scholarship Council—Hebrew University of Jerusalem Scholarship Program.

## Author contributions

X.W., R.N. and L.C. conceived the study. X.W. and R.N. developed the methods, analysed the data, and wrote the paper. R.N. conceived the MSSM method. L.C. and R.N. obtained funding. X.W., L.C., L.G., C.M., D.C., Y.Z., O.M., Z.X., N.B., A.K., H.P.J. and J.M. acquired animal movement tracking data. A.D.F. and L.G. provided important input during manuscript preparation, and all other authors commented and contributed to the final version.

## Additional information

**Competing interests:** The authors declare no competing interests.

