## [peer review file · Nature Communications]

Reviewer #1 (Remarks to the Author):

The authors conducted a test of the green wave hypothesis in birds, which included multiple species (including multiple feeding guilds) that were replicated across populations and continents. The motivation for the author's analyses stems from the idea that other factors might drive migration timing in birds (e.g., day length), and those factors may be highly correlated with the progression of the green wave. Thus, there has not been true, unequivocal support for the green wave hypothesis. I like the authors' approach to testing the green wave hypothesis. They clearly developed a null model (or multiple statistical null models) of movement between winter range and breeding grounds, and examined whether empirical data supported the green wave hypothesis better than the null model. This is only rarely done in papers testing the green wave hypothesis. That being said, I find some of the logic the authors use to motivate their analysis and develop their hypothesis framework to be weak or confusing.

Let's say that migration timing indeed correlates better with day length than vegetation indices such as instantaneous rate of green-up. As the authors state on L50-51, this would support an alternative explanation to the green wave hypothesis for migration timing. However, I do not think that a correlation between migration timing and day length unequivocally refutes the green wave hypothesis. In fact, it may reveal the cue by which animals use to move in concert with changing seasonal vegetation. This is especially important for birds that migrate very long distances, as the local cues at one stopover may not correlate with the future conditions at the next stopover hundreds of KM to the north. Thus, birds must rely on day length to make decisions that lead them to arriving at locations when forage quality is at or near its peak in quality (i.e., following the green wave).

Furthermore, the authors do not provide a compelling alternative to the green wave hypothesis. They mention day length, weather, competition, and fat deposits as alternatives in the intro. But they are never developed very strongly. In addition, competition and weather are the only alternative explanations that might help explain the timing of migration. As I said before, day length could simply be the cue animals use to surf the green wave. And further, fat deposits could simply be a cue for these individuals to begin seeking out high quality food, and thus will want to surf the green wave northward as spring unfolds. Anyway, the bottom line here is that I don't believe the authors have developed a logical alternative to the green wave hypothesis to explain the timing of migration in this intro. Without such an alternative, their logic in their argument that the green wave hypothesis is not supported ubiquitously does not come across very strongly.

One final issue is that the discussion/conclusion of the manuscript is weak and does not highlight very well the main contributions of the manuscript. Most importantly, the authors conclude on L176-180 that further behavioral and spatio-temporal information needs to be collected to understand the proximate drivers of bird migration. To me, the authors need to do a better job developing/conceptualizing alternative hypotheses, and then suggest ways to collect data to test predictions of those hypotheses. As is, the authors conclude that researchers need to simply collect

more data to understand avian migration. This suggestion does not follow the scientific method. See other issues mentioned below on lines 188-194.

Line by line comments (some very important as well)

L7-8. Based on the first sentence of the abstract, the authors assume that tracking seasonally changing resources is the main driver of animal migration. This may be true for avian migration, but other plausible (and supported) hypotheses exist for mammals such as escaping predation and insect harassment. See Avgar et al. 2013 (CJZ) for a review. Thus, I would suggest that the authors tone down their broad statements, and make sure they are specific to avian taxa when appropriate. Note that I am OK with the wording on L41-43.

L9-10. This sentence in the abstract is rather vague. Any way to be more specific here?

L15-19. Tense changes in this sentence. And this sentence is rather long.

L20. Could remove the words "be invoked to"

L20. "explain migration" is rather vague. Do the authors mean "explain the timing of migration" or "explain the evolution of migration"? It is unclear.

L19-24. OK, but what is the alternative (or null) hypothesis that was supported more than the green-wave hypothesis? Without this relative comparison, the relative strength of the results are tricky to interpret.

L24-25. This final sentence is not strong, and does not highlight a strong and tangible contribution to the literature. In fact, I don't even think it is a complete sentence. It seems that the authors need to add "that avian migration is" after the word 'suggest'.

L30. Fryxell and Avgar 2012 do not talk about avian taxa in their paper. Further, their paper is simply a review of Bischof et al. 2012, and thus I suggest to the authors that Bischof et al. 2012 be cited instead of Fryxell and Avgar. Also, between Bischof et al. 2012 and Merkle et al. 2016, the forage maturation hypothesis is clearly linked to the green-wave hypothesis for migratory mammalian herbivores. Thus, I don't think the authors should make such a clear distinction between green wave hypothesis for birds and forage maturation hypothesis for ungulates.

L49. Saying 'and other environmental factors' is rather vague. The authors need to be more specific so the reader can understand the logic in the author's argument about the alternatives to the green wave hypothesis.

L53-55. The authors should review Appendix 7 in Aikens et al. 2017. They do indeed examine the correlations found between the green wave and movements with a null model has some similarities to the author's ST, SS, and STS methods.

L59-62. The authors cannot make inference on 'all migratory herbivores' (as they state), only avian herbivores because of the data they have. Also, avian must be mentioned on lines 69-70 too.

L62-64. I don't think this statement is an 'alternative' to the green-wave hypothesis. It is simply better defining the green wave hypothesis to be specific to grazers.

L75-81. Usually when predictions are listed like this (e.g., P1, P2, P3) they all fall under the umbrella of a single hypothesis. Here, however, prediction 1 would support the green wave hypothesis, prediction 2 would support a modification of the green wave hypothesis to be specific to grazers, and prediction 3 would support an alternative to the green wave hypothesis that is not defined very well. I think the authors need to work on this section to better communicate their logic behind their hypotheses/predictions.

L83-86. There needs to be more info reported about these methods in the main text. Otherwise the reader must go to the methods at this point to even obtain a general idea of what SCC, CSM, and MSSM mean. Most importantly, brief definitions of stochastic migrations and metric selection must be added here.

L87-98. I like the logic behind the author's criteria here. In fact, these are all good predictions that would support the green wave hypothesis. However, the authors state "These criteria are most related our second prediction that only grazers follow the green wave" on L98-99, which confuses me as to what the authors are trying to convey with these criteria.

L113. Be specific with the type of data generated here. As I was reading, I thought suddenly that the authors were generating a different type of data than mentioned in previous sentence. Further confusing me is that I still have no idea what stochastic simulations are at this point in the manuscript.

L121-124. These two sentences confused me. Perhaps a conceptual diagram explaining what SCC and the other methods (including SS, ST, and STS) would help.

L127-128. I have no idea what the authors are reporting here. What models? What are the response and predictor variables here? Very unclear unless the reader reads the methods.

L139-140. This sentence stating that bill morphology did not explain green-wave surfing seems redundant from above, and it makes me think there is some tightening to do in the results section.

L141-142. This statement in regards to CSM "One should carefully consider its further application to test the green wave hypothesis" confuses me. Why are the authors using a method that they conclude should be carefully considered in order to test the green wave hypothesis?

L144-149. Again, this information is so difficult to interpret without pouring over the methods section. Any way to simplify so the manuscript can be understood without reading the methods (which comes after the main body) first?

L163-167. Authors should provide more info than just "geographical region also emerged as a strong predictor..." In other words, what regions were there green wave surfers and what regions were there not?

L168-169. This statement "Nevertheless, the significant effect of geographical area suggests that some regionally variable factors, other than the green wave, shape migration patterns of avian herbivores." needs much more discussion behind it. The authors need to develop their logic behind this conclusion, and outline some possible/plausible explanations. The authors mention this again on L188-191, but still do not develop their argument very strongly or convincingly.

L191-194. The authors end their manuscript with a mention of the methodology that they developed. I think there are other more important contributions that would much more strongly end this manuscript.

L229-231. How much error was involved in manually georeferencing stopover locations from printed maps? Perhaps the authors could have asked the owners of those data to determine those locations.

L236. What do the authors mean by 'sub-optimal' when talking about refueling sites? What is optimal anyway? peak green-up? If the authors are using sub-optimal as a way to describe the forage quality at a given stopover, then it is unclear to me how the green-wave hypothesis fits into the authors' conceptualization of migration. In other words, the green wave hypothesis is about herbivores visiting sites when at peak forage quality, and now the authors are removing stopovers that aren't at peak forage quality? I guess I simply don't understand.

L251. A 15km radius is a huge area! I think avian herbivores can chose foraging sites at a much finer scale than a 707 square kilometers, even if these species migrate in a large group. I think the authors should consider some type of sensitivity analysis to the radius size. Also, the authors say that their radius is 'moderate', but then they compare it to another 15 km radius and a 50 km radius, which seems confusing to me. Where do the authors get 'moderate' from?

L309-310. Do the authors mean 1,000 simulations per individual? Not clear.

L387. Must be a mistake here, or I am getting confused by the writing. The authors say that they 'drew stopover durations from the observed migration start dates'

L389-390. The number of acronyms is making it difficult to follow this manuscript. There are 5 acronyms just in this sentence!

Reviewer #2 (Remarks to the Author):

This manuscript presents a welcome and broad evaluation of the green wave hypothesis in avian herbivores – how migrants follow the spring green-up during their northward travel from wintering to breeding grounds. The authors investigate the relationship between several vegetation indices derived from the NDVI and spring migration tracked for 14 populations of 9 species of waterfowl. In addition to simple correlations between predicted and observed timing, the authors also apply a range of simulations to create null hypotheses against which to evaluate observed patterns. The authors find only limited support for the green wave hypothesis. Only in three of 14 populations do birds experience better instantaneous rate of green-up than random simulations, and in none for other indices. The authors conclude that tracking the northward advancement in foliage quality is not a general phenomenon. I find the study approach novel and most interesting, the methods appropriate and the conclusions generally justified. However, I identified several issues that I urge the authors to consider and I generally found the presentation overly complicated limiting the accessibility of the results.

I fully agree that the comparison of observations against simulated patterns as null models is a most useful approach for initial evaluation of the green wave hypothesis. If observed patterns are not significantly different from simulations then explanations are likely to be simple. The green wave hypothesis has generally be evaluated based on some very basic assumptions and I feel that the approach suffers from being slightly too simplified to actually support the more general conclusions. Obviously, NDVI is a very broad and crude vegetation measure – in reality birds might only exploit

certain parts of the vegetation and NDVI is not necessarily describing what the birds are benefitting from, and the actual relationship between NDVI-derived green-up indices and profitability might differ from those assumed (GWI, IRG). The evaluation of several metrics rather than just 50%GWI or IRG goes some way to meet this goal but at least this uncertainty should be acknowledged and discussed. An alternative would be to find the best relationship with the different measures (acknowledging that the exact relationship is not necessarily straightforward) and then test these models against random simulations.

Somewhat related to this issue, the authors argue (L. 50-55) that testing the validity of the green wave hypothesis requires going beyond mere correlations. I, of course, agree but simulation are also not really a tool for establishing causality – causality requires experimental approaches or natural experiments – but if as in this case observations do not differ from random patterns it is reasonable to conclude that one does not need to consider the more advanced mechanisms.

Regarding the simulations, I find it most useful with the randomisations in space and time separately as well as combined. However, I would like the authors to clarify and justify why stochastic tracks including the spatial component are modelled as continuous-time correlated random walks. I assume that simulated tracks according to this model will be biased toward the observed routes and thus have an overrepresentation of routes close to the observed one. It appears to me that random points (perhaps restricted to forward movement and a bond on total distance) within the potential migration range would be a simpler (and thus preferred) null model.

In my view, the presentation is overly complicated. There is a bewildering array of acronyms that the reader has to keep track of to understand what and how simulations have been done, and sentences are long and complicated. I would suggest the authors only presented the correlations and what they call their MSSM approach (without giving it an actual name). Their SCM approach is nested within MSSM and it would be fine to just note that this approach has been used in other studies and that no significant results were obtained based on 50% GWI. Also, an overview table of the approaches/scenarios simulated might be helpful for the reader.

In addition, the structure of the manuscript is somewhat unconventional with a very short discussion compared to the longer introduction - some more discussion about limitations of the approach and differences from other studies would be interesting here. The text seems to revert several times from slightly different angles to similar points leaving an impression of a slightly repetitive text. In general, the presentation of what concepts are actually tested was confusing – is following the seasonally changing vegetation a distal/proximate/ultimate/fundamental drivers/factors.

Kasper Thorup

Reviewer #3 (Remarks to the Author):

This is a timely and engaging study that tests the quality of the green wave hypothesis for a collection of herbivorous migratory bird species based on tracking information compiled during spring migration from across the globe. This study moves beyond species specific studies to test the generality of the green wave hypothesis in a novel fashion using a set of migration simulations. Below I outline some concerns I identified when reading the manuscript. I hope the authors find these suggestions to be useful.

1) I found the use of acronyms in the manuscript to be excessive, and some of the acronyms appear to be undefined. For example, "GW" and "GWH" are not defined, and "GWH" can be easily confused with "GWI". Acronyms help to condense the text and support more rapid comprehension, but in this case the prevalence of acronyms primarily generate confusion, as least for me. I would suggest only using acronyms when absolutely necessary, and if including multiple acronyms, providing a table with definitions.

2) In addition to the use of acronyms, the writing in the manuscript was not always very clear, especially in the Methods. I believe the manuscript would benefit by the authors editing the entire manuscript with the goal of promoting clarity and comprehension.

3) The stochastic migration simulations would benefit by a conceptual figure that describes how each is parametrized and designed. Currently, extracting this information from the Methods alone is challenging.

4) I would suggest a greater emphasis is needed throughout the manuscript on the fact this is a spring migration study.

5) The results from the analysis are summarized in one figure (Fig. 2) that lacks any information on how "green waves" are structured across space or by species. I would suggest adding maps to the manuscript that summarize how patterns of greenness are defined globally during spring migration based on the remote sensing data used in the study. I would also suggest adding figures to the manuscript that summarize the results of the double-logistic regression analyses for each species. The combination of greenness maps and species specific plots would add critical details to the results, which would enhance the depth and quality of the findings and conclusions.

6) The Discussion section is one paragraph in length, which is a little too concise, and the conclusion are prone to exaggeration. I would suggest expanding and refining the Discussion by first reviewing the recent green wave literature, including studies that use non-tracking data resources (e.g., eBird and radar) and studies that provide more theoretical assessments, and determining how this study's finding build upon this previous work. I would also suggest using the maps and figures I recommended above to help expand, refine, and balance the interpretation and conclusions presented in the Discussion.

Title: The phrase “seasonally changing vegetation” strikes me as much too general. It suggests that both spring and autumn migration will be considered and even suggests changes in vegetation outside of migration might be considered. A more informative title might be similar to: “Testing the ubiquity of the green wave hypothesis during spring migration for avian herbivores”

Fig. 1: Not critical, but it seems that it would be valuable to include scientific names in the figure or legend.

Fig. 2: The four columns in each panel are Obs, ST, SS and STS? This should be clarified in the legend.

Line 172: What is a “foliage quality wave”? I would suggest sticking to the same terminology throughout. Also, I do not think NDVI can be easily interpreted as “foliage quality.”

Lines 172-176: The conclusions presented here are prone to exaggeration. I would suggest a more careful assessment of the underlying green wave patterns and associations (through the addition of green wave maps and species specific plots) are needed when interpreting the analysis. I think his information will provide a more balanced and nuanced set of conclusions.

Line 181: There are other data resources in addition to tracking data that have been used to explore the green wave hypothesis. This includes citizen-science data such as eBird (see La Sorte et al. 2014 PRSB) and weather surveillance radar data (see work by Horton, Kelly, Bridge). I would suggest developing these points further in the Discussion. Also, recent work by Marius Somveille might also be relevant.

Lines 186-188: It is not clear to me what “local” is referring to here, and I could not find support for this statement in Kelly et al. 2016. It seems the issue is one of scale and how migrants sample the environment during migration. In this study, the sample resolution is 15 km radius circles. I would not classify this as “local.” There is much room to explore issues of scale in the Discussion.

Lines 259: What is the advantage of using a double-logistic model when modeling the green wave? Outlining the rationale here would be useful.

Line 277: Equation (c) is not very informative. Why not display the entire model?

Lines 254-283: I think it would be helpful to expand and clarify the rationale behind the methods described in this section. Right now, the reader has to refer to several citations to build an understanding of the logic behind the methods. It seems it would be valuable to introduce some of these details directly into the Methods.

Lines 275: The phrase “population of species” is a little confusing. Is this actually a collection of individuals for each species, and the mixed model is being applied to each species with individual nested in year included as a random effect?

Reviewer #4 (Remarks to the Author):

This work provides an assessment of whether the migration strategies of several species of waterbirds shows evidence for the currently popular “green wave hypothesis” (GWH). The novelty of this paper lies in its broad scale (multiple locations, multiple populations) and its extensive usage of simulations to test hypotheses. The findings indicate support for the GWH in only two species (3 populations) of grazing birds, and the authors conclude that tracking of foliage quality wave is not a global phenomenon nor a pervasive driver of migration of avian herbivores.

As a reviewer outside this immediate field, but involved in researching the large scale migration of marine megafauna, I enjoyed this paper and expect it would be of interest to a wide readership across a range of research fields. The issue is topical within the specific field as evidenced by recent related papers in high impact journals (Science Adv., Ecol. Lett., Curr. Biol.). Since support for the GWH remains mixed across papers, these findings ought to be highly cited and influence thinking in the evolving field.

Overall, I would like to see a revised version of this study published in Nature Communications. However, I do have a number of substantive comments that I would like to see addressed that particularly relate to the reporting of the data and methodology, and the presentation and interpretation of results. As can happen with large datasets and complex multi-stage methodologies many details are missing or require clarification. As requested by the editor I have mainly focussed on the methods/results in my review.

1) Reporting of data.

The authors should be commended for their efforts to collate a substantial dataset, comprising new empirical data, publicly available datasets and information from the literature. However these data are not comprehensively reported and some extra information is required that is both relevant to their analysis and that will aid readers outside of the field (who have no prior knowledge about

different bird tracking devices and the location information and error etc. obtained). In particular, up until L332 in the Methods it would seem that all tracking data of interest are reduced to summaries of the stopover locations; stopover durations; and flight segments between. Subsequently, the extensive use of the crawl package to simulate migration trajectories (at a 2h timestep) requires substantially more data reporting.

Supplementary Table 1 needs to additionally report the type of location data obtained from each study – if this is GPS in all cases that is simple, but it may also include solar GPS, ARGOS, solar ARGOS etc. (NB. if different location data types are used, then information on the location error as supplied to crawl needs also to be documented somewhere). As well as reporting the summarised stopover information Supplementary Table 1 should also report the higher-level tracking summaries i.e., the total number of locations provided by each study; mean \pm SD no. of location fixes per day across individuals; mean \pm SD and range of tracking period (days) across individuals, mean \pm SD track distance etc.

Similarly, Supplementary Table 4 ought to supply the above information at the level of the individual bird i.e., the type of location data, total number of location fixes, average no. of location fixes per day, total duration of tracking period (days), total tracked distance etc. As the manuscript currently stands there is not enough information given on the movement data/track type for readers unfamiliar with these loggers.

2) Reporting of methodology.

To my reading these authors have undertaken a thoughtful analysis of a large dataset, and again they should be commended. However given the complex sequence of steps, I think a conceptual or flow diagram in the supplementary material would be a significant help to the reader in following the methodology through multiple stages. Likewise, a table matrix for the simulation setup would assist in clarifying which elements are held the same and which were randomised in each scheme. For example across the three schemes (ST, SS and STS), if I check the elements: number of stopovers, stopover location, start day, stopover duration and flight time at least two question marks arise (is flight time kept the same in ST? Is start day kept the same in SS?).

L273-283. Simple conventional correlations (SCC). These are important later (L389, Supplementary Tables 2, 3) for evaluating the probabilities of obtaining the SCC-classified surfing type (or better). I would therefore expect the linear mixed model fits to be presented graphically: a supplemental figure is necessary showing the data points for Day_obs v Day_pred, and the population-level linear fit (&CI) together with the individual-level random slopes. This would be 14 panels total for all populations; with the slope coefficient and the p-values from Supplementary Table 2 printed on the figure panels. It is a little suspicious that the only 'perfect' surfers are the two data-poor populations (whooper swan and northern pintail) so it would be good for this data, and this method for designating 'partial' and 'perfect' surfers, to be plainly presented.

L314-331. While the challenges outlined are true enough, with respect to steps 1 and 2 of the four-step process I find myself wondering why - since the crawl package is being used - its inbuilt capacity for a "stopping model" was not utilised? With crwMLE the user can supply a stopping covariate (0 to 1) 'with 1 representing complete stop of the animal (no true movement, however, location error can still occur)'. This functionality was built for haulouts of marine mammals but is entirely analogous to bird migration stopovers; especially since the paper methods thus far show these stopovers have been pre-identified.

L330 & L377. The manuscript needs to report somewhere the total number of observed tracks per population (individual bird tracks (n) and the total number of tracking locations (N)) for which the 1000 simulated tracks are finally generated.

L397-402. MSSM – the primary conclusions of the paper rest mainly upon this approach, the results of which are presented in Fig. 2 (see related comment below). Instead of the multiple tests and post-hoc adjustments, it would seem more internally consistent with the rest of the paper to again adopt a LMM approach here. If migration experiment is treated as a factor as per Fig. 2 (Obs|ST|SS|STS) with Obs as the reference level, this would allow all other experiments (simulations) to be directly tested against the observations which is really the primary interest here. Within each experiment, we still have multiple measurements (green metric per stopover) per track (observed or simulated bird track) which ought to be accounted for via a random effects structure; but this is not currently the case. lme() further has the capacity to allow heterogeneous variance across the experiments, which seems clearly evident from the boxplots. I would suggest trying something along the lines of the following for each population/metric, and just checking the normalized residuals:

```
fit <-lme(GWindice ~ factor(MIGexperiment), random=~1 |  
factor(trackID),method="REML",  
weights=varIdent(form=~1 | factor(MIGexperiment)) )
```

3) Interpretation (and discussion) of results.

Keeping in mind the suggestion above for an alternate MSSM testing framework, I will comment nevertheless on the results presented in Fig. 2. The most systematic result evident is that in almost every case there is no difference detectable between the observed and simulated stochastic timing tracks (ST). This might suggest that birds are unable to obtain improved green metrics within the timing constraints of their migration schedule. However a caveat on this interpretation is the randomisation of stopover durations, since a more systematic manipulation of stopover timing (later or earlier) might yield a different result.

Related to this, it's evident in many cases that the Obs-ST pairs (most often paired a-a) are significantly different from the stochastic stopover site simulations, and importantly that better green indices are apparently obtainable should birds adopt alternate spatial trajectories and vary the spatial locations of their stopovers. While the authors report only weak differentiation between

feeding guilds, this second clear result has a very strong geographical trend, being apparent for four out of five East Asian populations for which stochastic stopover site simulations were available.

Taken together this seems to indicate inflexible (low) green indices along selected routes, but that birds largely do not opt to obtain potential benefits from altering flyways. The authors briefly allude to this in L187-191 of the Discussion, however the Discussion overall is extremely short and some expansion here is warranted. In particular the Current Biology paper by Yu et al (2017) appears relevant, which highlights the difference between geese species that will utilise intensive agricultural farmland adjacent to wetlands and those that confine stopovers to natural wetland habitats.

Yu, H., Wang, X., Cao, L., Zhang, L., Jia, Q., Lee, H., Xu, Z., Liu, G., Xu, W., Hu, B., Fox, A.D., 2017. Are declining populations of wild geese in China 'prisoners' of their natural habitats? *Curr Biol* 27, R376-R377.

The present study concludes support for the GWH in only three grazing populations located in Atlantic Europe (Barnacle geese from Svalbard and the Barents Sea; and the greater white fronted goose also from the Barents Sea). In light of the distinctly different results suggested between geographical regions (L163, 168), i.e., Atlantic Europe and East Asia, some further presentation and interpretation of results in an entirely spatial context would be valuable. For example, via summary seasonal maps over the two regions showing a relevant green metric such as IRG.

My final comment is that the general conclusion commencing the Discussion (L172-180) needs to be moderated within the context of the current study, i.e., limited to the interrogated scale of stopover scheduling. This paper does not address related concepts (e.g., 'jumpers' rather than 'surfers'); and a note in the Discussion about optimally timing arrival at the northern breeding site (e.g., Si et al, 2015) is probably also warranted.

Si, Y., Xin, Q., De Boer, W.F., Gong, P., Ydenberg, R.C., Prins, H.H., 2015. Do Arctic breeding geese track or overtake a green wave during spring migration? *Scientific reports* 5, 8749.

Specific comments

L22-23. Although the bill morphology association is acknowledged in the Results L151 its high-performance "goose-like" significance does not appear in the Discussion or elsewhere, only in this Abstract?

L28-33. This definition of the GWH seems to need something that more explicitly links migratory movements with quality resources being acquired en route. As it presently reads it could be understood by a non-specialist to simply refer to the large scale movement between two distinct seasonal ranges.

L75-81 (also L99-103). Although this provides a general conceptual context within which the results may be examined, L298-301 better captures the explicit hypotheses that are being tested in this study. A generalised form of the latter would be more valuable up front in the Introduction.

L89. I find the usage of “baseline criteria” here confusing. Could this be reworded to something like “we evaluated the results obtained by these methods in light of two general expectations”?

L110 and Supplementary Table 1. “Bird years” is not defined, and its relevance is unclear?

L177. “Three continents”. While the author’s efforts to compile data from the literature is admirable, the northern pintail information from N. America unfortunately provided very little to the study.

L185. Is “outside” meant to be “inside”?

L206. Data retrieved from Movebank should be explicitly noted somewhere – perhaps with a footnote attached to the “Published tracks” entries in Supplementary Table 1.

L227-31. Stopover/migration information. It seems disjointed that much later at L401 it is mentioned some populations only have observed and stochastic timing simulated migrations. These two populations (whooper swan and northern pintail) and their data limitations (and hence limited analyses) should be mentioned directly here where the data explanation is made.

L254. Report mean±SD number of pixels contributing per stopover location.

L258-268. A supplementary figure showing an example of the seasonal progression would be helpful for interpretation of this section, with time along the x-axis and the 3 vegetation metrics (NVDI, IRG and GWI) along the y-axis. Perhaps this could be developed for selected stopover locations across a couple of the major geographical regions (e.g., Barents Sea, E. Asia; which ultimately do/don’t show support for the GWH)?

L281. “..., or lower >1, and ...” – fulfilling this criteria would mean birds arrive much later than predicted but still with a significant relationship ($P < 0.05$) to GWI. This does not occur in the observations; does it occur in the simulations or is it unnecessary?

L291. In a paper with so many practical applications of modelling (statistical and simulation), is the choice of wording “null model” here (also L53) necessary? Aren’t you simply building “expected distributions” (via simulation) to compare the observations to?

L345. To ensure more directed movement at the migration start, and more variable movement towards the migration end?

L355-359. This sentence needs reworking – do you mean the net displacement (distance moved start to end) was within 200km of the observed net displacement?

L375-377. The meaning of this sentence is unclear, and needs some rewording.

L397. i.e., less than just expected by chance?

L421/L436. Would inclusion of feeding guild as a fixed effect be more in line with the overarching manuscript hypotheses (noting that biological family is not retained in any of the preferred models)?

L448. Strongly encourage authors to deposit their extensive compiled datasets into a public repository as a valuable legacy.

L666. Feeding guild classification is actually given in Supplementary Table 1.

Fig. 1. The number given in this legend for species, grazers, facultative herbivores do not match the manuscript text.

Supplementary Table 1. Geographical range would also be a helpful column entry in this Table, to provide the broader context for the Capture Location. In this Table it is not clear why Capture Location and Date are missing for 6 Mallards?

Supplementary Table 2. For each population, give the number of individuals (N) and the number of stopovers (n) included in the SCC mixed model. To match the text explanation (L280) Supplementary Table 2 should give the upper and lower 95% confidence limits (rather than the 95%CI). It would also be useful to report the variance estimate for the individual level random effects as lack of significance at the population level can be related to high individual-level variability.

I find the CSM result of so many probabilities being exactly 1.00 very surprising – i.e., every simulated track providing an equivalent or better fit to the GWH – are the authors entirely sure about these simulation results?

Supplementary Table 3. The presentation in this Table is difficult to interpret. It seems that a '+' under biological group indicates this variable was included; does the presence of a number (coefficient?) in the Upper mandible column indicate this variable was included? It would be more straightforward to include the model formulae. If all combinations of no variables (intercept only)/single variable/both variables are modelled this should be stated; and if so it seems the results for SCC upper mandible only are missing. Is the † in the correct place?

Supplementary Table 5. The parameter definitions need to be given (i.e., sigma = velocity variation, beta = velocity autocorrelation etc.). This Table heading would benefit from a brief expansion of how the drift and velocity autocorrelation and variance parameters may be interpreted (i.e., further model details; L345, L355).

Supplementary Figure 2. "...for all the five grazing populations" – aren't there six (pink-footed goose from Svalbard also)? Caption needs to state which simulation scheme results are presented – STS? It would also be helpful to add the relevant polygons from Supp. Fig. 3 that are used to spatially constrain the simulated tracks (L368).

RESPONSE TO REFEREES

Response 1 (a general response to major comments of the reviewers): We thank the four reviewers for your useful and constructive feedback on our manuscript NCOMMS-18-14699-T: *Testing the ubiquity of seasonally changing vegetation as the driver of avian herbivore migrations*. We addressed all comments as detailed below. The main changes include:

- (1) Clearer study focus. We emphasized in the manuscript that this study aims to test robustly the green wave hypothesis, and provided possible explanations to the results.
- (2) Improved description of the methods. We consulted the author of the *crawl* package, the core modelling tool of this study, to confirm our study methods. We added new tables and figures to illustrate our approach to test the green wave hypothesis and to generate stochastic migrations.
- (3) Reclassification of species and foraging guilds. We split the previous Bean Goose to two species, namely, Taiga Bean Goose and Tundra Bean Goose, according to the latest World Bird List (<http://www.worldbirdnames.org/>). The bill morphology data remained the same as in the original version, since they were based on the two different morphologies. We also reclassified Pink-footed Geese as facultative herbivores (previously grazer), as suggested by the coauthor who contributed the Pink-footed Goose tracking data and worked on this species for many years.
- (4) Better presentation of results. We reorganized Table 1 and Figure 2 to convey a straightforward presentation and comparison of different methods.
- (5) Extensively improved Discussion. In the new, completely re-written Discussion, we now discuss (1) the robustness of the main conclusion, (2) more in-depth interpretation and explanations of the findings, (3) comparisons with other works, (4) some practical implications and suggestions for future work.
- (6) Significant improvement of data presentation and readability. We provided more

details of the data in tables. We also followed the suggestions of the reviewers to reduce the extensive use of acronyms in the manuscript. See Responses 34 and 39 for details.

Our revision notes below describe the changes in detail.

Reviewer #1:

The authors conducted a test of the green wave hypothesis in birds, which included multiple species (including multiple feeding guilds) that were replicated across populations and continents. The motivation for the author's analyses stems from the idea that other factors might drive migration timing in birds (e.g., day length), and those factors may be highly correlated with the progression of the green wave. Thus, there has not been true, unequivocal support for the green wave hypothesis. I like the authors' approach to testing the green wave hypothesis. They clearly developed a null model (or multiple statistical null models) of movement between winter range and breeding grounds, and examined whether empirical data supported the green wave hypothesis better than the null model. This is only rarely done in papers testing the green wave hypothesis. That being said, I find some of the logic the authors use to motivate their analysis and develop their hypothesis framework to be weak or confusing.

Response 2: Thanks for emphasizing this important merit of the work, and for your constructive comments! To better convey the results and comparisons of different methods, we reorganized Table 1 and Fig. 2.

Let's say that migration timing indeed correlates better with day length than vegetation indices such as instantaneous rate of green-up. As the authors state on L50-51, this would support an alternative explanation to the green wave hypothesis for migration timing. However, I do not think that a correlation between migration timing and day length unequivocally refutes the green wave

hypothesis. In fact, it may reveal the cue by which animals use to move in concert with changing seasonal vegetation. This is especially important for birds that migrate very long distances, as the local cues at one stopover may not correlate with the future conditions at the next stopover hundreds of KM to the north. Thus, birds must rely on day length to make decisions that lead them to arriving at locations when forage quality is at or near its peak in quality (i.e., following the green wave).

Furthermore, the authors do not provide a compelling alternative to the green wave hypothesis. They mention day length, weather, competition, and fat deposits as alternatives in the intro. But they are never developed very strongly. In addition, competition and weather are the only alternative explanations that might help explain the timing of migration. As I said before, day length could simply be the cue animals use to surf the green wave. And further, fat deposits could simply be a cue for these individuals to begin seeking out high quality food, and thus will want to surf the green wave northward as spring unfolds. Anyway, the bottom line here is that I don't believe the authors have developed a logical alternative to the green wave hypothesis to explain the timing of migration in this intro. Without such an alternative, their logic in their argument that the green wave hypothesis is not supported ubiquitously does not come across very strongly.

Response 3: We thank the reviewer for highlighting this issue. In the Introduction, we argued that a significant correlation of migration progress with day length (or temperature, wind or other environmental factors) could not be taken as a strong evidence against the green wave hypothesis. Yes, we agree that day length could, in essence, serve as a cue for tracking changes in foliage quality, hence a significant correlation with day length could in fact *support* the green wave hypothesis. We raised this issue in the Introduction to make the point that inference from correlations is problematic, just because of such multiple (and often unknown) links that cannot be differentiated by correlations alone. This is exactly why we developed the null model approach that the reviewer liked. This approach is testing whether the observed

northward progress of migrating birds better match changes in green wave metrics compared to simulated stochastic northward tracks that were generated irrespective of the green wave. Although this approach does not suggest alternative mechanisms, it does provide a strong test for the green wave hypothesis. Please note that the main goal of our work was not to suggest alternative hypotheses for the green wave, but to test the ubiquity of the green wave hypothesis across different bird species, feeding guilds and populations, as reflected in the manuscript title and the main text (e.g. L25-28). **Nevertheless, our results do suggest that human disturbance might explain this lack of agreement with the green wave hypothesis.** As detailed in Responses 69, we now elaborate this alternative explanation in a new paragraph in the Discussion dedicated to this topic (L223-249) as well as further emphasis in the end of the Abstract (L25-28).

Overall, following this comment, we have changed the MS:

- (1) to avoid confusion about the study aims (L52-54,70-71),
- (2) to clarify that we do provide a strong test for the green wave hypothesis (L72-76), and
- (3) to propose human disturbance as an alternative hypothesis accounting for the deviations between the green wave expectations and the observed migration patterns (L25-28, 223-249).

One final issue is that the discussion/conclusion of the manuscript is weak and does not highlight very well the main contributions of the manuscript. Most importantly, the authors conclude on L176-180 that further behavioral and spatio-temporal information needs to be collected to understand the proximate drivers of bird migration. To me, the authors need to do a better job developing/conceptualizing alternative hypotheses, and then suggest ways to collect data to test predictions of those hypotheses. As is, the authors conclude that researchers need to simply collect more data to understand avian migration. This suggestion does not follow the scientific method. See other issues mentioned below on lines 188-194.

Response 4: As explained in Responses 1 and 3, we totally revised the Discussion to properly highlight the main contributions of the manuscript. In this work, we aim to provide a strong test to the green wave hypothesis using an advanced statistical approach as well as a large and diverse dataset, which mark two important novel merits of this work. We are confident that the rather weak support we found for the green wave hypothesis would motivate further research to formulate and test alternative hypotheses. We agree that some potential hypotheses could be further developed based on our findings, but this must be done with caution, since the current study was not designed to test any alternative hypothesis. Nevertheless, as explained also in Response 3, in L223-249 we discussed one potential alternative explanation (geographical variation in human disturbance), which was only briefly mentioned in the previous version, but was born from our results and presumably bear strong effect on bird migration. In L328-331 we outlined ways to test this alternative hypothesis as the reviewer suggested. We also enriched the Discussion following suggestions of Reviewer 4 (see Responses 67-71), which nicely fit this alternative human disturbance hypothesis.

Line by line comments (some very important as well)

L7-8. Based on the first sentence of the abstract, the authors assume that tracking seasonally changing resources is the main driver of animal migration. This may be true for avian migration, but other plausible (and supported) hypotheses exist for mammals such as escaping predation and insect harassment. See Avgar et al. 2013 (CJZ) for a review. Thus, I would suggest that the authors tone down their broad statements, and make sure they are specific to avian taxa when appropriate. Note that I am OK with the wording on L41-43.

Response 5: We agree and reworded this sentence.

L9-10. This sentence in the abstract is rather vague. Any way to be more specific here?

Response 6: We rephrased and shortened this sentence to make it more specific.

L15-19. Tense changes in this sentence. And this sentence is rather long.

Response 7: We made the tenses consistent and shortened the sentence.

L20. Could remove the words “be invoked to”

Response 8: Done.

L20. “explain migration” is rather vague. Do the authors mean “explain the timing of migration” or “explain the evolution of migration”? It is unclear.

Response 9: We changed ‘migration’ to ‘migration progress’.

L19-24. OK, but what is the alternative (or null) hypothesis that was supported more than the green-wave hypothesis? Without this relative comparison, the relative strength of the results are tricky to interpret.

Response 10: The null hypothesis is that migrating birds migrate north irrespective of the green wave, see L21-22. Re the alternative hypotheses, please see our clarifications in Responses 3 and 4.

L24-25. This final sentence is not strong, and does not highlight a strong and tangible contribution to the literature. In fact, I don’t even think it is a complete sentence. It seems that the authors need to add “that avian migration is” after the word ‘suggest’.

Response 11: We rewrote the whole sentence.

L30. Fryxell and Avgar 2012 do not talk about avian taxa in their paper. Further, their paper is simply a review of Bischof et al. 2012, and thus I suggest to the authors that Bischof et al. 2012 be cited instead of Fryxell and Avgar. Also, between Bischof et al. 2012 and Merkle et al. 2016, the forage maturation hypothesis is clearly linked to the green-wave hypothesis for migratory mammalian herbivores. Thus, I don’t think the authors should make such a clear

distinction between green wave hypothesis for birds and forage maturation hypothesis for ungulates.

Response 12: It is indeed more appropriate to cite Bischof et al. 2012 instead of Fryxell and Avgar (2012). We rephrased the statement of green wave hypothesis and foraging maturation hypothesis in the sentence (L32-33).

L49. Saying ‘and other environmental factors’ is rather vague. The authors need to be more specific so the reader can understand the logic in the author’s argument about the alternatives to the green wave hypothesis.

Response 13: We provided some examples of ‘other factors’ in this sentence (L52).

L53-55. The authors should review Appendix 7 in Aikens et al. 2017. They do indeed examine the correlations found between the green wave and movements with a null model has some similarities to the author’s ST, SS, and STS methods.

Response 14: It is true that Aikens et al. (2017) used a null model approach to test whether animal migration coincided with green wave metrics, which resembles our stochastic stopover simulations (change only in space). Their work was explicitly acknowledged in L85 of the previous version, and we kept this acknowledgement in the revised one as well (L95). We are well aware of the method used in Aikens’ paper (their Appendix 7): they randomly sampled movement segments between the summering and wintering ranges to assign stochastic movement tracks, while preserving the daily rate of displacement. They explicitly admit that their method is rather coarse. We therefore further refined the stochastic simulations by changing time and not only space (adding two temporal versions of stochastic simulations), and preserving some other characteristics of the movement tracks in the spatial only simulation type, as detailed in the new Supplementary Table 2. The differences between the approach taken by Aikens et al. and our approach and are now further clarified in L285-288.

L59-62. The authors cannot make inference on ‘all migratory herbivores’ (as they

state), only avian herbivores because of the data they have. Also, avian must be mentioned on lines 69-70 too.

Response 15: Done.

L62-64. I don't think this statement is an 'alternative' to the green-wave hypothesis. It is simply better defining the green wave hypothesis to be specific to grazers.

Response 16: We agree and replaced the word 'alternatively' to 'or more specifically', and 'furthermore' to 'Alternatively'.

L75-81. Usually when predictions are listed like this (e.g., P1, P2, P3) they all fall under the umbrella of a single hypothesis. Here, however, prediction 1 would support the green wave hypothesis, prediction 2 would support a modification of the green wave hypothesis to be specific to grazers, and prediction 3 would support an alternative to the green wave hypothesis that is not defined very well. I think the authors need to work on this section to better communicate their logic behind their hypotheses/predictions.

Response 17: We apologize for the insufficient explanations of the logic behind our predictions. We now modified the text (L81-92) to clarify that (1) these are different predictions pertaining to the level of support one might expect for the green wave hypothesis, they are *not* predictions related to different/alternative hypotheses (see Response 3 for further explanations), and (2) that the logic behind the three predictions refers to a decrease in the level of support across species and populations, in the following way:

P1 – the green wave will be supported in *all* population of *all* species

P2 – the green wave will be supported in *all* populations of *some* species (grazers)

P3 – the green wave will be supported in *some* population of *some* species (grazers)

L83-86. There needs to be more info reported about these methods in the main text. Otherwise the reader must go to the methods at this point to even obtain a

general idea of what SCC, CSM, and MSSM mean. Most importantly, brief definitions of stochastic migrations and metric selection must be added here.

Response 18: We added Supplementary Figs. 4 & 5 and Supplementary Tables 1 & 2 to explain the brief stochastic simulations and metric selection. See also Responses 21, 39, 43, 44 and 62.

L87-98. I like the logic behind the author's criteria here. In fact, these are all good predictions that would support the green wave hypothesis. However, the authors state "These criteria are most related our second prediction that only grazers follow the green wave" on L98-99, which confuses me as to what the authors are trying to convey with these criteria.

Response 19: We hope that our clarification re the three predictions (Response 17) will also clarify this confusion.

L113. Be specific with the type of data generated here. As I was reading, I thought suddenly that the authors were generating a different type of data than mentioned in previous sentence. Further confusing me is that I still have no idea what stochastic simulations are at this point in the manuscript.

Response 20: Thanks for this suggestion; "data" was indeed inappropriate here. We changed 'data' to 'stochastic migrations' in this sentence. We also added a brief description of the simulations (See Response 18).

L121-124. These two sentences confused me. Perhaps a conceptual diagram explaining what SCC and the other methods (including SS, ST, and STS) would help.

Response 21: We added the new Supplementary Tables 1 & 2.

L127-128. I have no idea what the authors are reporting here. What models? What are the response and predictor variables here? Very unclear unless the reader reads the methods.

Response 22: We apologize for this confusion. These are the statistical models, as we now clarify in L179-181.

L139-140. This sentence stating that bill morphology did not explain green-wave surfing seems redundant from above, and it makes me think there is some tightening to do in the results section.

Response 23: We rephrased these sentences to clarify their meaning (L152-153).

L141-142. This statement in regards to CSM “One should carefully consider its further application to test the green wave hypothesis” confuses me. Why are the authors using a method that they conclude should be carefully considered in order to test the green wave hypothesis?

Response 24: We maintained the SCC since it has been used intensively in previous studies of the green wave hypothesis (see references in L95). Furthermore, a simple version of the CSM was used in Aikens et al. 2017 (see Response 14). It was therefore important to test these existing methods, rather than to exclude them in advance. As we stated in this particular sentence and in L153-155, our findings raise doubts about the utility of these two methods, and we believe that this is an important message for future studies of this hypothesis. Furthermore, the superior performance of the MSSM method introduced here cannot be justified without an adequate comparison with other methods.

L144-149. Again, this information is so difficult to interpret without pouring over the methods section. Any way to simplify so the manuscript can be understood without reading the methods (which comes after the main body) first?

Response 25: We now provided further explanation of this in the Introduction section, see L97-107.

L163-167. Authors should provide more info than just “geographical region also emerged as a strong predictor...” In other words, what regions were there green

wave surfers and what regions were there not?

Response 26: Information added (L185-190). See also Response 3.

L168-169. This statement “Nevertheless, the significant effect of geographical area suggests that some regionally variable factors, other than the green wave, shape migration patterns of avian herbivores.” needs much more discussion behind it. The authors need to develop their logic behind this conclusion, and outline some possible/plausible explanations. The authors mention this again on L188-191, but still do not develop their argument very strongly or convincingly.

Response 27: We agree and now fully discuss this particular result as described in Responses 3, 4, 40 and 69.

L191-194. The authors end their manuscript with a mention of the methodology that they developed. I think there are other more important contributions that would much more strongly end this manuscript.

Response 28: We have totally revised the Discussion, please see Responses 1, 3 and 4.

L229-231. How much error was involved in manually georeferencing stopover locations from printed maps? Perhaps the authors could have asked the owners of those data to determine those locations.

Response 29: Only two datasets analyzed in this MS were derived from literature. For one of them (the whooper swan study) the track coordinates were available in the publication, but not for the other one (the northern pintail study). The authors of this paper were contacted but did not reply. However, their paper included names of lakes and valleys, and maps with locations, providing sufficient information from which we were able to determine the coordinates. The robust results of the sensitivity analyses of buffer size from 5 km to 30 km (Supplementary Table 6, Response 31) suggest that the conclusions of the analyses are not sensitive to the error in determining locations from literature.

L236. What do the authors mean by 'sub-optimal' when talking about refueling sites? What is optimal anyway? peak green-up? If the authors are using sub-optimal as a way to describe the forage quality at a given stopover, then it is unclear to me how the green-wave hypothesis fits into the authors' conceptualization of migration. In other words, the green wave hypothesis is about herbivores visiting sites when at peak forage quality, and now the authors are removing stopovers that aren't at peak forage quality? I guess I simply don't understand.

Response 30: We agree. We changed the threshold for stopover duration to 48 hours (L381-384), as other studies do, and updated stopover information (Supplementary Table 3). All results are now based on the 48-hour threshold. Using this new threshold did not change our conclusions (Supplementary Table 6).

L251. A 15km radius is a huge area! I think avian herbivores can chose foraging sites at a much finer scale than a 707 square kilometers, even if these species migrate in a large group. I think the authors should consider some type of sensitivity analysis to the radius size. Also, the authors say that their radius is 'moderate', but then they compare it to another 15 km radius and a 50 km radius, which seems confusing to me. Where do the authors get 'moderate' from?

Response 31: We conducted sensitivity analyses of buffer size ranging from 5 km to 30 km and found that all main conclusions of the work were unchanged (Supplementary Table 6). Nevertheless, although a very small radius might seem preferable, it might not capture the typical size of a stopover site hence raise uncertainty on green wave metrics representing the foraging conditions in this site. The max displacement during stopover was in the order of 10-20 km (see added column in Supplementary Table 3) and therefore we decided to presented the main results using a radius of 5-km (rather than 15 km in the previous version), which is still smaller than previous studies.

L309-310. Do the authors mean 1,000 simulations per individual? Not clear.

Response 32: Yes, per individual. Changed to ‘per individual bird’ (L457).

L387. Must be a mistake here, or I am getting confused by the writing. The authors say that they ‘drew stopover durations from the observed migration start dates’

Response 33: Sentence corrected.

L389-390. The number of acronyms is making it difficult to follow this manuscript. There are 5 acronyms just in this sentence!

Response 34: Sentence re-written. See also Response 39.

Reviewer #2 (Signed by Kasper Thorup):

This manuscript presents a welcome and broad evaluation of the green wave hypothesis in avian herbivores – how migrants follow the spring green-up during their northward travel from wintering to breeding grounds. The authors investigate the relationship between several vegetation indices derived from the NDVI and spring migration tracked for 14 populations of 9 species of waterfowl. In addition to simple correlations between predicted and observed timing, the authors also apply a range of simulations to create null hypotheses against which to evaluate observed patterns. The authors find only limited support for the green wave hypothesis. Only in three of 14 populations do birds experience better instantaneous rate of green-up than random simulations, and in none for other indices. The authors conclude that tracking the northward advancement in foliage quality is not a general phenomenon. I find the study approach novel and most interesting, the methods appropriate and the conclusions generally justified. However, I identified several issues that I urge the authors to consider and I generally found the presentation overly complicated limiting the accessibility of

the results.

Response 35: Many thanks for your kind words and your useful comments and suggestions. To better convey the results and comparisons of different methods, we reorganized Table 1 and Fig. 2.

I fully agree that the comparison of observations against simulated patterns as null models is a most useful approach for initial evaluation of the green wave hypothesis. If observed patterns are not significantly different from simulations then explanations are likely to be simple. The green wave hypothesis has generally be evaluated based on some very basic assumptions and I feel that the approach suffers from being slightly too simplified to actually support the more general conclusions. Obviously, NDVI is a very broad and crude vegetation measure – in reality birds might only exploit certain parts of the vegetation and NDVI is not necessarily describing what the birds are benefitting from, and the actual relationship between NDVI-derived green-up indices and profitability might differ from those assumed (GWI, IRG). The evaluation of several metrics rather than just 50%GWI or IRG goes some way to meet this goal but at least this uncertainty should be acknowledged and discussed. An alternative would be to find the best relationship with the different measures (acknowledging that the exact relationship is not necessarily straightforward) and then test these models against random simulations.

Response 36: We agree and added further emphasis in L294-300 stating (1) that each index has its pros and cons and none of the indices can capture fine details such as which parts of the vegetation are being eaten by the birds, and (2) that the use of multiple indices that collectively represent a variety of vegetation effects better than any single index alone, partially resolves this problem, as the reviewer explained. We considered the alternative method suggested by the reviewer, but decided not to adopt it since it indeed requires making (strong) assumptions about unknown relationships between the indices, and the conclusions, in turn, would largely depend on these assumptions.

Somewhat related to this issue, the authors argue (L. 50-55) that testing the validity of the green wave hypothesis requires going beyond mere correlations. I, of course, agree but simulation are also not really a tool for establishing causality – causality requires experimental approaches or natural experiments – but if as in this case observations do not differ from random patterns it is reasonable to conclude that one does not need to consider the more advanced mechanisms.

Response 37: Yes, this comment goes in line with the logic we developed in this manuscript. We simulated stochastic migrations to test if the observed migration patterns are different from random northward migration irrespective of the green wave. In most cases we failed to find statistically significant differences, therefore we suggested (e.g., in L19-22) that the green wave hypothesis does not provide a better explanation compared to random northward progress, in agreement with the reviewer's comment. We also added a note (L311-314) that experimental manipulations or natural experiments are more difficult to perform, and are usually rather restricted in their temporal or spatial extent, hence interpretation is also limited for these approaches.

Regarding the simulations, I find it most useful with the randomisations in space and time separately as well as combined. However, I would like the authors to clarify and justify why stochastic tracks including the spatial component are modelled as continuous-time correlated random walks. I assume that simulated tracks according to this model will be biased toward the observed routes and thus have an overrepresentation of routes close to the observed one. It appears to me that random points (perhaps restricted to forward movement and a bond on total distance) within the potential migration range would be a simpler (and thus preferred) null model.

Response 38: As we now further explain in L499&502-507, the random migrations were simulated to resemble the observed migration patterns to maintain the movement characteristics of each individual. Our approach accommodates variation among

individuals that cannot be kept by the approach suggested by the reviewer (and also used by Aikens et al. 2017, see Response 14). For example, stochastic simulations of migration of a bird migrating in constant speed between A and B and a bird migrating in highly variable speed (with the same mean) between the same two points should be different.

In my view, the presentation is overly complicated. There is a bewildering array of acronyms that the reader has to keep track of to understand what and how simulations have been done, and sentences are long and complicated. I would suggest the authors only presented the correlations and what they call their MSSM approach (without giving it an actual name). Their SCM approach is nested within MSSM and it would be fine to just note that this approach has been used in other studies and that no significant results were obtained based on 50% GWI. Also, an overview table of the approaches/scenarios simulated might be helpful for the reader.

Response 39: We apologize for the complicated presentation. To address this problem, we made the following improvements:

- (1) we removed all acronyms and kept just one (MSSM) and other established green wave anonyms (NDVI, GWI and IRG), since they are very commonly used throughout the paper;
- (2) we added descriptions of each method (SCC, CSM and MSSM) when mentioning them, in the last paragraph of Introduction (L97-107);
- (3) we added a new conceptual diagram to illustrate the simulation details (Supplementary Fig. 5, see also Response 18, 21, 44 and 62). However, we maintained the description and analyses of CSM to present a comprehensive analysis of different methods using various metrics and to justify our claims about the superior performance of the MSSM approach (see Response 24). The removed acronyms include:

SCC: Simple Conventional Correlation of arrival time,

CSM: Correlation method evaluated by simulating Stochastic Migrations,

GW: green wave,

GWH: green wave hypothesis,

ST: stochastic timing modelling,

SS: stochastic stopover site modelling, when migration tracks are available,

STS: stochastic timing and stopover site modelling, when migration tracks are available.

In addition, the structure of the manuscript is somewhat unconventional with a very short discussion compared to the longer introduction - some more discussion about limitations of the approach and differences from other studies would be interesting here. The text seems to revert several times from slightly different angles to similar points leaving an impression of a slightly repetitive text. In general, the presentation of what concepts are actually tested was confusing – is following the seasonally changing vegetation a distal/proximate/ultimate/fundamental drivers/factors.

Response 40: Thanks for these suggestions. We rewrote the entire Discussion to include (1) the main finding and its explanation, (2) more in-depth interpretation and explanations of the variable support of the green wave hypothesis, (3) comparisons with other works/methods, (4) broader implications and future directions. See also Response 1, 3, 4, 27 and 69.

Reviewer #3:

This is a timely and engaging study that tests the quality of the green wave hypothesis for a collection of herbivorous migratory bird species based on tracking information compiled during spring migration from across the globe. This study moves beyond species specific studies to test the generality of the green wave hypothesis in a novel fashion using a set of migration simulations. Below I

outline some concerns I identified when reading the manuscript. I hope the authors find these suggestions to be useful.

Response 41: We appreciate the reviewer's helpful and perceptive comments on the manuscript. They indeed improved the readability and focus of the manuscript.

1) I found the use of acronyms in the manuscript to be excessive, and some of the acronyms appear to be undefined. For example, "GW" and "GWH" are not defined, and "GWH" can be easily confused with "GWI". Acronyms help to condense the text and support more rapid comprehension, but in this case the prevalence of acronyms primarily generate confusion, as least for me. I would suggest only using acronyms when absolutely necessary, and if including multiple acronyms, providing a table with definitions.

Response 42: We reduced the use of most acronyms, as Response 39.

2) In addition to the use of acronyms, the writing in the manuscript was not always very clear, especially in the Methods. I believe the manuscript would benefit by the authors editing the entire manuscript with the goal of promoting clarity and comprehension.

Response 43: We followed this suggestion and thoroughly revised the whole manuscript including the Methods section. Major changes are the removal of most acronyms, new Supplementary Tables 1&2 and Supplementary Fig. 4. See also Responses 18, 21, 22, 39, 44, 62.

3) The stochastic migration simulations would benefit by a conceptual figure that describes how each is parametrized and designed. Currently, extracting this information from the Methods alone is challenging.

Response 44: We followed this suggestion and added a conceptual diagram explaining the migration simulations (Supplementary Fig. 5). See also Responses 21, 39 and 43.

4) I would suggest a greater emphasize is needed throughout the manuscript on

the fact this is a spring migration study.

Response 45: We changed the title to ‘Testing the ubiquity of green wave as the driver of spring migrations of avian herbivores’, and clarified this throughout the manuscript. See also Response 48.

5) The results from the analysis are summarized in one figure (Fig. 2) that lacks any information on how “green waves” are structured across space or by species. I would suggest adding maps to the manuscript that summarize how patterns of greenness are defined globally during spring migration based on the remote sensing data used in the study. I would also suggest adding figures to the manuscript that summarize the results of the double-logistic regression analyses for each species. The combination of greenness maps and species specific plots would add critical details to the results, which would enhance the depth and quality of the findings and conclusions.

Response 46: Many thanks for these valuable suggestions. We reorganized/added the two suggested figures to the Supplementary Material (Fig. 2& Supplementary Fig. 3). See also Responses 52, 63, 70.

6) The Discussion section is one paragraph in length, which is a little too concise, and the conclusion are prone to exaggeration. I would suggest expanding and refining the Discussion by first reviewing the recent green wave literature, including studies that use non-tracking data resources (e.g., eBird and radar) and studies that provide more theoretical assessments, and determining how this study’s finding build upon this previous work. I would also suggesting using the maps and figures I recommended above to help expand, refine, and balance the interpretation and conclusions presented in the Discussion.

Response 47: We thoroughly revised the Discussion section and provided information of related research in Discussion, as well as in Introduction (L35-43). See also Responses 4 and 40 to a similar comment from reviewers 1 and 2. For supporting figures, see our Response 46.

Title: The phrase “seasonally changing vegetation” strikes me as much too general. It suggests that both spring and autumn migration will be considered and even suggests changes in vegetation outside of migration might be considered. A more informative title might be similar to: “Testing the ubiquity of the green wave hypothesis during spring migration for avian herbivores”

Response 48: See Response 45.

Fig. 1: Not critical, but it seems that it would be valuable to include scientific names in the figure or legend.

Response 49: Done.

Fig. 2: The four columns in each panel are Obs, ST, SS and STS? This should be clarified in the legend.

Response 50: We added descriptions of the four acronyms in the Fig. 2 and Supplementary Fig. 2 captions.

Line 172: What is a “foliage quality wave”? I would suggest sticking to the same terminology throughout. Also, I do not think NDVI can be easily interpreted as “foliage quality.”

Response 51: We changed ‘foliage quality wave’ to ‘green wave’, and ‘migration’ to ‘spring migration’.

Lines 172-176: The conclusions presented here are prone to exaggeration. I would suggest a more careful assessment of the underlying green wave patterns and associations (through the addition of green wave maps and species specific plots) are needed when interpreting the analysis. I think his information will provide a more balanced and nuanced set of conclusions.

Response 52: Regarding the green wave maps and species-specific plots, these were added following the reviewer’s suggestion, please see Response 46. These additions,

however, do not change our results and conclusions. As explained in our Response 3, two of the main merits of our approach are:

- (1) the use of stochastic simulations to provide a stronger test for the green wave hypothesis compared to the previous correlation-based studies, and
- (2) the use of migration tracking data from multiple species, populations and individuals, providing a more comprehensive analysis than all previous studies thus far.

These merits have been emphasized by Reviewer 2 (text above Response 35) and Reviewer 4 (text above Response 59). We now better clarify these points, please see Responses 4 and 70, for example. Overall, we maintain that our analyses do provide strong evidence that the green wave hypothesis is not a robust general mechanism that determines spring migration of birds, as stated in these lines.

Line 181: There are other data resources in addition to tracking data that have been used to explore the green wave hypothesis. This includes citizen-science data such as eBird (see La Sorte et al. 2014 PRSB) and weather surveillance radar data (see work by Horton, Kelly, Bridge). I would suggesting developing these points further in the Discussion. Also, recent work by Marius Somveille might also be relevant.

Response 53: These are indeed important complementary approaches to study bird migration but the very coarse spatial resolution and the lack of information at the individual bird level that are inherent in such data sources rather restrict our ability to test the green wave hypothesis. We have previously cited a relevant publication of this group (Kelly et al. 2016, Bridge et al. 2016), and now added citation to the relevant study of La Sorte et al. (2014, Proc B) too. We now discuss the methodological pros and cons of these and other approaches, see L310-320.

Lines 186-188: It is not clear to me what “local” is referring to here, and I could not find support for this statement in Kelly et al. 2016. It seems the issue is one of scale and how migrants sample the environment during migration. In this study,

the sample resolution is 15 km radius circles. I would not classify this as “local.”

There is much room to explore issues of scale in the Discussion.

Response 54: The “local” scale in Kelly et al. 2016 was a radius of 100 km, which is an order of magnitude larger than the scale we used. We added this clarification and discussed the scale issue in L300-309.

Lines 259: What is the advantage of using a double-logistic model when modeling the green wave? Outlining the rationale here would be useful.

Response 55: This model has the best performance in a comparison of NDVI filtering approaches, and widely used in migration - green wave studies. We now explain the merits of using this model in L416-417.

Line 277: Equation (c) is not very informative. Why not display the entire model?

Response 56: We agree and changed equation (c) to represent the entire model.

Lines 254-283: I think it would be helpful to expand and clarify the rationale behind the methods described in this section. Right now, the reader has to refer to several citations to build an understanding of the logic behind the methods. It seems it would be valuable to introduce some of these details directly into the Methods.

Response 57: We now visually portray the differences between these metrics in Supplementary Fig. 4.

Lines 275: The phrase “population of species” is a little confusing. Is this actually a collection of individuals for each species, and the mixed model is being applied to each species with individual nested in year included as a random effect?

Response 58: No, we run a separate model for each population of each species. Populations are defined in L364-365, 573-575. We removed the words ‘of species’ to avoid further confusion.

Reviewer #4 (Remarks to the Author):

This work provides an assessment of whether the migration strategies of several species of waterbirds shows evidence for the currently popular “green wave hypothesis” (GWH). The novelty of this paper lies in its broad scale (multiple locations, multiple populations) and its extensive usage of simulations to test hypotheses. The findings indicate support for the GWH in only two species (3 populations) of grazing birds, and the authors conclude that tracking of foliage quality wave is not a global phenomenon nor a pervasive driver of migration of avian herbivores.

As a reviewer outside this immediate field, but involved in researching the large scale migration of marine megafauna, I enjoyed this paper and expect it would be of interest to a wide readership across a range of research fields. The issue is topical within the specific field as evidenced by recent related papers in high impact journals (Science Adv., Ecol. Lett., Curr. Biol.). Since support for the GWH remains mixed across papers, these findings ought to be highly cited and influence thinking in the evolving field.

Overall, I would like to see a revised version of this study published in Nature Communications. However, I do have a number of substantive comments that I would like to see addressed that particularly relate to the reporting of the data and methodology, and the presentation and interpretation of results. As can happen with large datasets and complex multi-stage methodologies many details are missing or require clarification. As requested by the editor I have mainly focussed on the methods/results in my review.

Response 59: Thanks very much for this positive feedback and excellent suggestions.

1) Reporting of data.

The authors should be commended for their efforts to collate a substantial dataset, comprising new empirical data, publicly available datasets and information from the literature. However these data are not comprehensively reported and some extra information is required that is both relevant to their analysis and that will aid readers outside of the field (who have no prior knowledge about different bird tracking devices and the location information and error etc. obtained). In particular, up until L332 in the Methods it would seem that all tracking data of interest are reduced to summaries of the stopover locations; stopover durations; and flight segments between. Subsequently, the extensive use of the crawl package to simulate migration trajectories (at a 2h timestep) requires substantially more data reporting.

Supplementary Table 1 needs to additionally report the type of location data obtained from each study – if this is GPS in all cases that is simple, but it may also include solar GPS, ARGOS, solar ARGOS etc. (NB. if different location data types are used, then information on the location error as supplied to crawl needs also to be documented somewhere). As well as reporting the summarised stopover information Supplementary Table 1 should also report the higher-level tracking summaries i.e., the total number of locations provided by each study; mean \pm SD no. of location fixes per day across individuals; mean \pm SD and range of tracking period (days) across individuals, mean \pm SD track distance etc.

Response 60: We agree and have modified the new Supplementary Table 3 (previously Supplementary Table 1) accordingly. Thank you!

Similarly, Supplementary Table 4 ought to supply the above information at the level of the individual bird i.e., the type of location data, total number of location fixes, average no. of location fixes per day, total duration of tracking period (days), total tracked distance etc. As the manuscript currently stands there is not enough information given on the movement data/track type for readers unfamiliar with these loggers.

Response 61: Ditto (now Supplementary Table 7).

2) Reporting of methodology.

To my reading these authors have undertaken a thoughtful analysis of a large dataset, and again they should be commended. However given the complex sequence of steps, I think a conceptual or flow diagram in the supplementary material would be a significant help to the reader in following the methodology through multiple stages. Likewise, a table matrix for the simulation setup would assist in clarifying which elements are held the same and which were randomised in each scheme. For example across the three schemes (ST, SS and STS), if I check the elements: number of stopovers, stopover location, start day, stopover duration and flight time at least two question marks arise (is flight time kept the same in ST? Is start day kept the same in SS?).

Response 62: Done (now Supplementary Fig. 5). See also Response 18, 21 and 44.

L273-283. Simple conventional correlations (SCC). These are important later (L389, Supplementary Tables 2, 3) for evaluating the probabilities of obtaining the SCC-classified surfing type (or better). I would therefore expect the linear mixed model fits to be presented graphically: a supplemental figure is necessary showing the data points for Day_obs v Day_pred, and the population-level linear fit (&CI) together with the individual-level random slopes. This would be 14 panels total for all populations; with the slope coefficient and the p-values from Supplementary Table 2 printed on the figure panels. It is a little suspicious that the only 'perfect' surfers are the two data-poor populations (whooper swan and northern pintail) so it would be good for this data, and this method for designating 'partial' and 'perfect' surfers, to be plainly presented.

Response 63: Excellent suggestions, many thanks. We reorganized Fig. 2 to present the SCC for each population.

L314-331. While the challenges outlined are true enough, with respect to steps 1

and 2 of the four-step process I find myself wondering why - since the crawl package is being used - its inbuilt capacity for a “stopping model” was not utilised? With crwMLE the user can supply a stopping covariate (0 to 1) ‘with 1 representing complete stop of the animal (no true movement, however, location error can still occur)’. This functionality was built for haulouts of marine mammals but is entirely analogous to bird migration stopovers; especially since the paper methods thus far show these stopovers have been pre-identified.

Response 64: Thanks for this suggestion too. We consulted the author of the crawl package, Dr. Devin Johnson, and according to his advice, our approach (which is based on separation of stopover and flight phase) is, in principle, equally legitimate to using crwMLE of analyzing the whole track (with all stopovers and flights) while adding a stopping covariate. We discussed the two approaches with Dr Johnson, and shared the conclusion that our approach is more appropriate for migrating birds, for three major reasons. First, in order to incorporate a ‘stopping covariate’ into the model, we have to first designate its value, because tracking data of terrestrial migrating birds typically lack a signal for “haulouts”, as opposed to the dry sensor customarily used in marine mammals. To do this, we can either set its value in the whole stopover period as ‘1’ (full stopping), or set its value based on the day-night rhythm. However, the former will consider any within-stopover-site movement, which can be >20 km per hour as error; the latter will be very arbitrary because of the lack of information of their behavior. Second, the movement modes between migratory flight and stopover are very different, and in essence, we should apply multi-state models to model the whole migration process if we do not knock out the stopover period. Finally, the advantage of incorporating a stopping covariate lies in the ability to investigate the movement within stopover period, which is not the interest of this study. Therefore, we maintained the current modeling process in the revised manuscript.

L330 & L377. The manuscript needs to report somewhere the total number of observed tracks per population (individual bird tracks (n) and the total number of tracking locations (N)) for which the 1000 simulated tracks are finally generated.

Response 65: Done, see Supplementary Table 3.

L397-402. MSSM – the primary conclusions of the paper rest mainly upon this approach, the results of which are presented in Fig. 2 (see related comment below). Instead of the multiple tests and post-hoc adjustments, it would seem more internally consistent with the rest of the paper to again adopt a LMM approach here. If migration experiment is treated as a factor as per Fig. 2 (Obs/ST/SS/STS) with Obs as the reference level, this would allow all other experiments (simulations) to be directly tested against the observations which is really the primary interest here. Within each experiment, we still have multiple measurements (green metric per stopover) per track (observed or simulated bird track) which ought to be accounted for via a random effects structure; but this is not currently the case. lme() further has the capacity to allow heterogeneous variance across the experiments, which seems clearly evident from the boxplots. I would suggest trying something along the lines of the following for each population/metric, and just checking the normalized residuals:

```
fit <-lme(GWindice ~ factor(MIGexperiment), random=~1/  
factor(trackID),method="REML",  
weights=varIdent(form=~1|factor(MIGexperiment)) )
```

Response 66: We tried a similar code before the first submission but failed to consistently normalize the residuals of the various metrics, populations etc. We have now applied the code kindly suggested by the reviewer and encountered the same problem. Inspired by this comment, we further explored other methods that can incorporate random/block effects in the model and are robust with residual normality and variance heterogeneity, such as Nemenyi's all-pairs comparisons tests of Friedman-type ranked data in PMCMRplus package and robust anova in WRS2 package. However, the former requires balanced design and the latter yielded out an error of the optimizer - 'too complex problem to solve'. We therefore regret that we cannot follow this piece of otherwise very good advice.

3) *Interpretation (and discussion) of results.*

Keeping in mind the suggestion above for an alternate MSSM testing framework, I will comment nevertheless on the results presented in Fig. 2. The most systematic result evident is that in almost every case there is no difference detectable between the observed and simulated stochastic timing tracks (ST). This might suggest that birds are unable to obtain improved green metrics within the timing constraints of their migration schedule. However a caveat on this interpretation is the randomisation of stopover durations, since a more systematic manipulation of stopover timing (later or earlier) might yield a different result.

Response 67: We thank the reviewer for this suggestion! To alleviate the randomization-related caveat mentioned by the reviewer, we examined whether migrating birds miss the green wave because they arrived to the stopover sites earlier or later than the green wave peak. We did this by calculating the difference (in days) between the observed arrival day to each stopover site and the expected peak green wave day at each site, for all birds, all populations and all species. These analyses are now presented in Supplementary Table 5. We found that six out of nine facultative herbivore and omnivore populations migrated significantly earlier than the green wave to the stopover site, while the time difference was not significant for all grazing populations, and all populations showed a large confidence interval. We now suggest that birds are not only constrained by timing of the migration schedule (as suggested by the reviewer), but also by the geographical pattern of human disturbance, as we now further discuss in the Discussion (L223-249). See also Responses 3 and 4.

Related to this, it's evident in many cases that the Obs-ST pairs (most often paired a-a) are significantly different from the stochastic stopover site simulations, and importantly that better green indices are apparently obtainable should birds adopt alternate spatial trajectories and vary the spatial locations of their stopovers. While the authors report only weak differentiation between feeding

guilds, this second clear result has a very strong geographical trend, being apparent for four out of five East Asian populations for which stochastic stopover site simulations were available.

Response 68: Very good point, thank you. We highlighted this trend in the Results (L190-194) and discussed it in the Discussion (L235-249). See also Responses 3 and 4.

Taken together this seems to indicate inflexible (low) green indices along selected routes, but that birds largely do not opt to obtain potential benefits from altering flyways. The authors briefly allude to this in L187-191 of the Discussion, however the Discussion overall is extremely short and some expansion here is warranted. In particular the Current Biology paper by Yu et al (2017) appears relevant, which highlights the difference between geese species that will utilise intensive agricultural farmland adjacent to wetlands and those that confine stopovers to natural wetland habitats.

Yu, H., Wang, X., Cao, L., Zhang, L., Jia, Q., Lee, H., Xu, Z., Liu, G., Xu, W., Hu, B., Fox, A.D., 2017. Are declining populations of wild geese in China ‘prisoners’ of their natural habitats? Curr Biol 27, R376-R377.

Response 69: We agree. We further extended the Discussion along these lines (L235-249) (see also Response 3), and added referral to Yu et al.

The present study concludes support for the GWH in only three grazing populations located in Atlantic Europe (Barnacle geese from Svalbard and the Barents Sea; and the greater white fronted goose also from the Barents Sea). In light of the distinctly different results suggested between geographical regions (L163, 168), i.e., Atlantic Europe and East Asia, some further presentation and interpretation of results in an entirely spatial context would be valuable. For example, via summary seasonal maps over the two regions showing a relevant green metric such as IRG.

Response 70: We revised and much extended our Discussion of the regional differences in migration-green wave association, please see Responses 3, 4, 27 and 40. We added seasonal maps of IRG and other metrics as also suggested by Reviewer 3, see the new Supplementary Fig. 5 and Response 46.

My final comment is that the general conclusion commencing the Discussion (L172-180) needs to be moderated within the context of the current study, i.e., limited to the interrogated scale of stopover scheduling. This paper does not address related concepts (e.g., ‘jumpers’ rather than ‘surfers’); and a note in the Discussion about optimally timing arrival at the northern breeding site (e.g., Si et al, 2015) is probably also warranted.

Si, Y., Xin, Q., De Boer, W.F., Gong, P., Ydenberg, R.C., Prins, H.H., 2015. Do Arctic breeding geese track or overtake a green wave during spring migration? Scientific reports 5, 8749.

Response 71: We agree and now discussed stopover timing in L238-239, 266-269, see also Response 67.

Specific comments

L22-23. Although the bill morphology association is acknowledged in the Results L151 its high-performance “goose-like” significance does not appear in the Discussion or elsewhere, only in this Abstract?

Response 72: We rephrased the sentences to better explain the feature of this bill shape, see L22-23, 211-213.

L28-33. This definition of the GWH seems to need something that more explicitly links migratory movements with quality resources being acquired en route. As it presently reads it could be understood by a non-specialist to simply refer to the large scale movement between two distinct seasonal ranges.

Response 73: We added “drive ‘the progression of’ migration” in L34 to avoid such

confusion.

L75-81 (also L99-103). Although this provides a general conceptual context within which the results may be examined, L298-301 better captures the explicit hypotheses that are being tested in this study. A generalised form of the latter would be more valuable up front in the Introduction.

Response 74: We agree. A true test of migration – green wave should consider both spatial and temporal aspects. We now further emphasize this point in L107-109.

L89. I find the usage of “baseline criteria” here confusing. Could this be reworded to something like “we evaluated the results obtained by these methods in light of two general expectations”?

Response 75: Thanks for the suggestion, and we reworded this sentence as suggested.

L110 and Supplementary Table 1. “Bird years” is not defined, and its relevance is unclear?

Response 76: We now define this term in L126-127.

L177. “Three continents”. While the author’s efforts to compile data from the literature is admirable, the northern pintail information from N. America unfortunately provided very little to the study.

Response 77: Indeed, North America was presented by only one population. Published tests of the the green wave hypothesis are rather scarce for migrating waterbirds of North America, and the North American studies mentioned in Response 53 are mostly relevant to passerines. Although movement data on migratory waterbirds have been collected, data availability limitations have precluded their incorporation in this manuscript (see a specific example in Response 29). We deleted the word “three” to avoid justified criticism of this sort, and added a note in the Discussion L320-323.

L185. Is “outside” meant to be “inside”?

Response 78: Corrected to ‘inside’.

L206. Data retrieved from Movebank should be explicitly noted somewhere – perhaps with a footnote attached to the “Published tracks” entries in Supplementary Table 1.

Response 79: All published tracks were retrieved from Movebank Data Repository, and each Movebank dataset is fully referenced in the Data source column of this Table.

L227-31. Stopover/migration information. It seems disjointed that much later at L401 it is mentioned some populations only have observed and stochastic timing simulated migrations. These two populations (whooper swan and northern pintail) and their data limitations (and hence limited analyses) should be mentioned directly here where the data explanation is made.

Response 80: We added a short paragraph in the end of this section to clarify that in these two cases we had stopover sites but not tracks, hence limited the simulations (discussed later on in L387-392).

L254. Report mean \pm SD number of pixels contributing per stopover location.

Response 81: Done, see L414-415.

L258-268. A supplementary figure showing an example of the seasonal progression would be helpful for interpretation of this section, with time along the x-axis and the 3 vegetation metrics (NVDI, IRG and GWI) along the y-axis. Perhaps this could be developed for selected stopover locations across a couple of the major geographical regions (e.g., Barents Sea, E. Asia; which ultimately do/don’t show support for the GWH)?

Response 82: We added an illustration (Supplementary Fig. 4) to differentiate the three indices (see Response 57), and global maps of indices’ dynamics

(Supplementary Fig. 3) (see Response 46, 52), to aid interpretation of this section.

L281. "..., or lower >1, and ..." – fulfilling this criteria would mean birds arrive much later than predicted but still with a significant relationship ($P < 0.05$) to GWI. This does not occur in the observations; does it occur in the simulations or is it unnecessary?

Response 83: Such cases are missing from the observations, and occur rather infrequently (0.3%) in the simulations. The main purpose of setting this criterion to provide a complete set of cases and to avoid confusion.

L291. In a paper with so many practical applications of modelling (statistical and simulation), is the choice of wording "null model" here (also L53) necessary? Aren't you simply building "expected distributions" (via simulation) to compare the observations to?

Response 84: We followed the standard definition of null models in ecology by Gotelli and Graves (1996, Null models in Ecology, Animal Behavior Society), as follows: "A null model is a pattern-generating model that is based on randomization of ecological data". This definition fits well our approach. We now cited this source in L58-59 and L458.

L345. To ensure more directed movement at the migration start, and more variable movement towards the migration end?

Response 85: This was implemented to preserve the general tendency of reduced autocorrelation with increasing time lag in animal tracks; that is, any two adjacent time points (not only in the start) tend to be more correlated with each other than two points far apart. We added this definition of the time lag to avoid such confusion (L513).

L355-359. This sentence needs reworking – do you mean the net displacement (distance moved start to end) was within 200km of the observed net

displacement?

Response 86: No, this is the displacement between the simulated end and the observed end. We revised this sentence to clarify this point (L524-525)

L375-377. The meaning of this sentence is unclear, and needs some rewording.

Response 87: Done.

L397. i.e., less than just expected by chance?

Response 88: We changed the method to completely follow Aikens et al. (2017), see L556-564) and Response 14.

L421/L436. Would inclusion of feeding guild as a fixed effect be more in line with the overarching manuscript hypotheses (noting that biological family is not retained in any of the preferred models)?

Response 89: We intentionally kept feeding guild out of this analysis, since we use this classification to test the level of support for our hypotheses, hence the results should be independent (as much as possible) from the evaluation criterion.

L448. Strongly encourage authors to deposit their extensive compiled datasets into a public repository as a valuable legacy.

Response 90: We followed the data availability policy of the journal.

L666. Feeding guild classification is actually given in Supplementary Table 1.

Response 91: This is true, but added here to make our main findings more visible to the readers.

Fig. 1. The number given in this legend for species, grazers, facultative herbivores do not match the manuscript text.

Response 92: We apologies for this error, and now report the corrected numbers.

Supplementary Table 1. Geographical range would also be a helpful column entry in this Table, to provide the broader context for the Capture Location. In this Table it is not clear why Capture Location and Date are missing for 6 Mallards?

Response 93: The table has been renumbered as Supplementary Table 7. The original Table included a geographical range column (fourth from the left). A geographical range column and the missing data for mallards were added in Supplementary Table 7.

Supplementary Table 2. For each population, give the number of individuals (N) and the number of stopovers (n) included in the SCC mixed model.

Response 94: The table has been renumbered as Supplementary Table 5. These numbers are reported for the observed tracks in Supplementary Table 1, and are the same in the SCC analysis, which is based on the observed data (see L432), hence adding this information seems redundant.

To match the text explanation (L280) Supplementary Table 2 should give the upper and lower 95% confidence limits (rather than the 95%CI).

Response 95: The table has been renumbered as Supplementary Table 5. Done.

It would also be useful to report the variance estimate for the individual level random effects as lack of significance at the population level can be related to high individual-level variability.

Response 96: Done.

I find the CSM result of so many probabilities being exactly 1.00 very surprising – i.e., every simulated track providing an equivalent or better fit to the GWH – are the authors entirely sure about these simulation results?

Response 97: Yes, these are the genuine results after rounding to two digits after the decimal point.

Supplementary Table 3. The presentation in this Table is difficult to interpret. It seems that a '+' under biological group indicates this variable was included; does the presence of a number (coefficient?) in the Upper mandible column indicate this variable was included? It would be more straightforward to include the model formulae. If all combinations of no variables (intercept only)/single variable/both variables are modelled this should be stated; and if so it seems the results for SCC upper mandible only are missing. Is the † in the correct place?

Response 98: Thank you for the suggestion. The table has been renumbered as Supplementary Table 4. We reorganized the table according to the suggestions and added text to the Table caption to clarify all these uncertainties.

Supplementary Table 5. The parameter definitions need to be given (i.e., sigma = velocity variation, beta = velocity autocorrelation etc.). This Table heading would benefit from a brief expansion of how the drift and velocity autocorrelation and variance parameters may be interpreted (i.e., further model details; L345, L355).

Response 99: The table has been renumbered as Supplementary Table 8. Done.

Supplementary Figure 2. "...for all the five grazing populations" – aren't there six (pink-footed goose from Svalbard also)?

Response 100: Corrected (now Supplementary Fig. 1).

Caption needs to state which simulation scheme results are presented – STS?

Response 101: This is relevant for both STS and SS, now clarified in the Figure caption.

It would also be helpful to add the relevant polygons from Supp. Fig. 3 that are used to spatially constrain the simulated tracks (L368).

Response 102: Done (the figure has been renumbered as Supplementary Fig. 1).

Reviewer #1 (Remarks to the Author):

The authors have improved their manuscript greatly. I applaud them for that. However, I still have one major concern and a few minor comments.

My major concern stems from my understanding of some of the logic in the discussion. The authors first conclude that "...only the MSSM approach met the general expectations of the green wave hypothesis, revealing that only a few grazer populations traced the green wave during spring migration." Then the authors conclude that "...tracking the green wave is neither a global phenomenon nor a pervasive driver of spring migration of avian herbivores." Yet, the authors go on to hypothesize that "human disturbance, which plays an important role in determining the progress of bird migration could explain the geographical deviations from the green wave hypothesis." My issue with these three conclusions is that if human disturbance is constraining birds from properly tracking the green wave, how can the authors conclude that one method is better than the other, and/or that the green wave hypothesis is not a global phenomenon that drives spring migration in birds? I am not sure how to fix this, but as is, I am having trouble understanding the broad takeaways from the paper.

Minor comments:

L70. What is meant by "diverse food types"? Could the authors be clearer here?

L102-103. This is not enough explanation of the MSSM method here. Please provide a bit more of an explanation of MSSM here so that the reader does not have to pour through the methods to understand.

L181-184. The authors say "total variance" here. Thus, it is necessary to state how much of the variance is unexplained by the model. Doing so puts the stated percentages in context.

L310-331. I feel like much of this paragraph is unnecessary for this paper.

Reviewer #3 (Remarks to the Author):

I appreciate the effort made by the authors to address my initial concerns with the study. The manuscript is much improved, and several of my previous concerns have been adequately addressed. However, there are a few issues that were not accounted for in this draft. One is the study's justification presented in the Introduction. The arguments provided here are not completely accurate or compelling, especially when discussing the use of null models and issue related to correlation. Null models provide a basis for making inferences on the likelihood of observed associations occurring by chance alone. However, null models do not provide a basis to assess the causal role for a single explanatory factor. If multiple explanatory factors were examined in combination, the relative strength of each could be tested. However, only the green wave hypothesis is tested in this study. The other factors listed here (latitude, day length, and temperature) do not provide the basis in this study for testing alternative hypotheses (the Abstract and Introduction suggest that this might actually be the case). Overall, I believe there is still room to improve and clarify the study's justification. Another issue is the presence of exaggerated statements. I appreciate the value of having a broad inferential scope, but each study must frame their inferences based on the data at hand. Globally there are around 180 species of water fowl, North America has around 40 species. Only one North American species was considered in this study. Statements in the manuscript that the data is taxonomically and geographically comprehensive are clear exaggerations. The manuscript should provide an objective assessment of what was actually measured and what can be inferred based on the data. I think it would be valuable to state directly in the manuscript that the sample used in this study represents around 6% of the world's water fowl species (this value should be confirmed) with the majority of the data coming from Europe and East Asia. I appreciate the effort made by the authors to expand the Discussion, but the exaggerated statements I identified in the previous version remain. Looking at the first paragraph of the Discussion, I would not describe the data as "exceptionally comprehensive" and I would argue that the study's findings do not support the conclusion that: "...tracking the green wave is neither a global phenomenon nor a pervasive driver of spring migration of avian herbivores." I would suggest taking a more objective look at the data and the results, and providing a more balanced and nuanced assessment of the study's findings that avoids absolute or extreme statements.

Reviewer #4 (Remarks to the Author):

I commend the work undertaken by the authors to revise their manuscript. Overall, I find it much improved particularly with respect to clarifications in the Introduction setting up the work, and the vastly expanded scope of the new Discussion. The expanded Supplementary material (flow diagram, simulation matrix) is also helpful in orienting the reader through a relatively complex work. In most cases the authors have worked to address the reviews, but in a few cases I believe a little more work is required.

The expanded discussion within the spatiotemporal context (L223-273) substantially enhances the storyline but there are components that still require some further development:

Responses 67 & 68. The additional text in the Results (L190-194) refer only to the issue of timing, and doesn't seem quite sufficient as is. I am not sure about the statement "This suggests that non-surfers timed their migration irrespective of the green wave". Doesn't it rather indicate they cannot obtain improved green metrics within the overall timing constraints of their migration schedule?

There are related Results elements that only appear in the Discussion (L234-239). These would more sensibly be located at the end of the Results, where they could be more clearly explained (especially the time differences of Suppl. Table 5) and better set the readers up for the ensuing Discussion.

The new text (Results and Discussion) particular to the East Asia migrants is welcome; but, please carefully note that the new results reported in the Fig 2 are not identical to previous, with the IRG results for observed East Asian migrants now being reported as "a-b" in several cases. The only clear separations (observed-simulated) using IRG are therefore for the GWFG and the tundra swan. The results do still hold however across East Asia species for the other metrics (GWI and NVDI; Suppl. Fig. 2). Hence the reporting, which ought to close off the Results section, would more correctly be something along the lines of:

"Our results for East Asia indicate that migrants simulated with stochastic stopover sites apparently follow the green wave better than the observed ones. In this flyway, the single grazer (the greater white-fronted goose) and all the facultative herbivores modelled using stochastic stopovers could apparently gain higher green wave index (GWI) values by stochastic migration (Fig. 2 and Suppl. Fig. 2). A similar result holds for the NVDI metric (excepting the bean goose) but is less clearly seen in the IRG."

This might be a good place to link to the new maps provided in Suppl. Fig. 3, which are not referred to yet that I can see. Perhaps zooming these to cover the main area of interest (Western Europe-Eastern Asia) might help tell us what is going on with the regional spatiotemporal dynamics, and assist in the Discussion development. Similarly, regarding the new Suppl. Figure 4, the request was to develop a couple of these time series for selected stopover sites for populations that do/do not support the green wave hypothesis, and therefore may be expected to be different (Eg in East Asia cf Barents Sea or Svalbard). A good example of something like this is in Fig. 3 of Aikens et al (2017); here, individual lines may be for birds within populations? Better spatial (SFig3) and temporal (SFig4) representations ought to help tie the story altogether. This issue was also specifically highlighted by Reviewer 3 (their point 5 - I would also suggest adding figures to the manuscript that summarize the results of the double-logistic regression analyses for each species. The combination of greenness maps and species specific plots would add critical details to the results, which would enhance the depth and quality of the findings and conclusions.) and remains only partially addressed.

Final point on the regional variation (L250-273). Two species not deeply discussed – the Scandinavian bean goose and Svalbard pink footed goose – now appear to show the exact opposite pattern to what is described for the East Asian migrants. That is, observed migrants follow the green wave better than simulated migrants with stochastic stopovers (spatial simulations, new results

using IRG, Fig 2). Referring back to the paper's framework at L465-468: is this then partial support for the green wave hypothesis, i.e., only spatial selectivity for sites with a green wave? c.f. statements at L165, L188, L283 "no significant association". And if so, are the results for these species relevant to the Discussion about when/how species have capacity to adapt the timing of their ocean crossings versus their ability to select optimal sites en route?

Other comments:

L218-224. Awkward transition between paragraphs since in both cases ("while another population" and "a notable deviation") the same population is being discussed (East Asian GWFG); needs streamlining.

L271. A good place to refer to approaches that systematically manipulate the timing of migration schedules (forwards or backwards in time) rather than the randomisation approach adopted here, for example the space-time-time matrix of Bischof et al (2012) *American Naturalist* <https://www.journals.uchicago.edu/doi/10.1086/667590>.

L298. The utility of multiple metrics seems rather lost in the current version.

L367-8 "fewer than five individuals or up to ten stopover sites" is confusing.

L392. Name the relevant bird populations missing data.

Supplementary Table 2, Examples. Preface "Assume" with something like "Evaluating the simple method using stochastic migrations, assume..." and follow "therefore it is a green wave surfer" with "under CSM". Similarly, preface "When the chance" with something like "Using the Metric Selection approach, when ..."

Supplementary Table 2. Fill all cells for absolute clarity (NAs if not applicable).

Supplementary Figure 1. Add "Examples are shown for the STS scheme".

Response 62. The LMM addition to Figure 2 is welcome, although Panels A and B now need their axes explained in the caption. It would still be good to see the full mixed effects model results in a supplementary figure, with a panel per population, showing the population fit (as per Figure 2A) but also the individual-level random slopes. As per the new variance information in Supplementary Table 5 there is very high variance at the individual bird level in some populations.

Response 64. I appreciate the authors efforts here, though I am surprised by the cited of movement >20km/h at stopovers, since this seems inconsistent with the 5-30km buffers (L405) set at stopover sites for the extraction of green-wave metrics?

Response 66. Literally did you try: `fit <- lme(...as described); E <- resid(fit, type="normalized")`? Also a log transform on the metrics prior could help? If the authors retain their multiple tests approach in favour of an lme approach, somewhere in the manuscript there needs to be an explicit caveat about the unaccounted for autocorrelation (as per my original comment @L397-402).

A couple of remaining queries on the dataset:

L341: three sources? Only two stated.

L342 “movement information first presented in this study” – and Supplementary Table 3 – are any of the GPS tracks from Eastern China–Eastern Russia birds (GWFG, Swan goose, Bean goose) previously published in the Yu et al (2017) Current Biology paper?

Responses 65 and 94: the manuscript text makes reference to exclusion of tracks based on various criteria (eg L366-71; L498). As Supplementary Table 3 currently reads, it looks like this reports the “raw” tracking data, prior to this processing. If it in fact represents all the tracking data to which the SCC is applied, and from which simulations are generated, this could be stated explicitly in the table heading. For example with a sentence like: “Data from a total of 193 birds (222 migration tracks) comprising a total of xxx GPS locations were retained for use in the correlation analysis and to generate stochastic simulations (n = 1000 each scheme, see Supplementary xxx)”.

Data availability: a statement confirming that all relevant data are available from the authors is the minimum data availability requirement for this journal. “Upon reasonable request” makes it sound like the authors may have some fuzzy (unstated) restrictions? If so these need to be clearly stated.

On a very general note, I suggest choosing a more specific terminology than “avian herbivores” in the title and throughout. Given that the study species are ducks, geese and swans, perhaps migratory waterfowl? Or herbivorous waterbirds?

Similarly, the tone could be moderated throughout as some new sections have a tendency to over-emphasise the point. A few examples here, but the authors should screen the text: L11 “failed” could be “struggle”. L17 and L197 “exceptionally comprehensive” – although 193 birds is admirable, tracking studies are never going to manage to be “comprehensive” relative to population sizes

(unless possibly in very tragically endangered cases). Moderate to “large” or “substantial”? Change L297 “robust” to read “conservative”; L290 “superior” to read “reliable” etc.

RESPONSE TO REFEREES

Response 1 (a general response to major comments of the reviewers): We thank the three reviewers for your extremely useful and constructive feedback on our manuscript NCOMMS-18-14699A: *Testing the ubiquity of the green wave as the driver of spring migration of herbivorous waterfowl*. We addressed all comments as detailed below. The main changes include:

- (1) Clearer study justification. In addition to further clarification in the manuscript of the use of the null model approach, we added a new analysis in the revised manuscript to test air temperature as an additional alternative explanation for spring migrations. This analysis further strengthens our conclusions.
- (2) Significantly improved presentation of results. We added the new Supplementary Fig. 2 to include the migration and its associated greenscape, as well as the double-logistic curve results, for *all* study populations. We also added the new Supplementary Fig. 3 to show both population-level and individual-level patterns of green-wave surfing using the Simple Conventional Correlation method.
- (3) Moderated tone over the manuscript to avoid exaggerated or over-emphasized statements.

Our revision notes below describe the changes in detail.

Reviewer #1:

The authors have improved their manuscript greatly. I applaud them for that.

However, I still have one major concern and a few minor comments.

Response 2: Thanks for your kind words and constructive comments!

My major concern stems my understanding of some of the logic in the discussion.

The authors first conclude that "...only the MSSM approach met the general

expectations of the green wave hypothesis, revealing that only a few grazer populations traced the green wave during spring migration.” Then the authors conclude that “...tracking the green wave is neither a global phenomenon nor a pervasive driver of spring migration of avian herbivores.” Yet, the authors go on to hypothesize that “human disturbance, which plays an important role in determining the progress of bird migration could explain the geographical deviations from the green wave hypothesis.” My issue with these three conclusions is that if human disturbance is constraining birds from properly tracking the green wave, how can the authors conclude that one method is better than the other, and/or that the green wave hypothesis is not a global phenomenon that drives spring migration in birds? I am not sure how to fix this, but as is, I am having trouble understanding the broad takeaways from the paper.

Response 3: We considered the two issues raised by the reviewer in a hierarchical two-step manner. In the first step, we evaluated the relative performance of three alternative methods using two independent evaluation criteria for testing the green wave hypothesis. Based on the results of the best performing method (MSSM), we concluded that the green wave is not a global driver waterbird spring migration, as reflected by the lack of support for most species/populations included in our analysis. This conclusion contrasted the wide and growing recognition of the green wave hypothesis based on single species studies, motivating us to further explore why this hypothesis was not supported for most species in our study. This, and geographical bias observed in our results, led us to explore human disturbance as a potential explanation for deviations from the green wave in the second step of our inquiry. We reported here some evidence supporting the human disturbance hypothesis. This support for the human disturbance hypothesis does not contradict the lack of support to the green wave hypothesis, but rather explains the observed deviations from the green wave hypothesis. To avoid further confusion of this kind, we carefully checked the entire revised manuscript and changed the wording in L219-222.

Minor comments:

L70. What is meant by “diverse food types”? Could the authors be clearer here?

Response 4: Even grazers can sometime exploit non-leaf or non-plant material. We clarified this in L69-70.

L102-103. This is not enough explanation of the MSSM method here. Please provide a bit more of an explanation of MSSM here so that the reader does not have to pour through the methods to understand.

Response 5: Done (L102-110).

L181-184. The authors say “total variance” here. Thus, it is necessary to state how much of the variance is unexplained by the model. Doing so puts the stated percentages in context.

Response 6: Done (L195).

L310-331. I feel like much of this paragraph is unnecessary for this paper.

Response 7: In addition to satellite/GPS tracking, many other methods have been used to test the migration – green wave associations, including citizen science data, radar and field manipulations. In the previous round of reviews, one of the reviewers urged us to provide a short overview of these other methods (see Response 53 of the previous round), an addition we found both appropriate and worthy in itself, and as a basis for discussing the pros and cons of our methods and alternative ones, as clarified in L358-363.

Reviewer #3 (Remarks to the Author):

I appreciate the effort made by the authors to address my initial concerns with the study. The manuscript is much improved, and several of my previous concerns have been adequately addressed. However, there are a few issues that were not accounted for in this draft. One is the study’s justification presented in the

Introduction. The arguments provided here are not completely accurate or compelling, especially when discussing the use of null models and issue related to correlation. Null models provide a basis for making inferences on the likelihood of observed associations occurring by chance alone. However, null models do not provide a basis to assess the causal role for a single explanatory factor. If multiple explanatory factors were examined in combination, the relative strength of each could be tested. However, only the green wave hypothesis is tested in this study. The other factors listed here (latitude, day length, and temperature) do not provide the basis in this study for testing alternative hypotheses (the Abstract and Introduction suggest that this might actually be the case). Overall, I believe there is still room to improve and clarify the study's justification.

Response 8: We greatly appreciate your comments and feedback. We used stochastic simulations to test the green wave hypothesis by comparing the observed tracks against stochastic **northward** movement irrespective of the green wave, as stated in L53-57, 78-81 of the manuscript. Therefore, our basic null model does consider latitude, or more precisely northward latitudinal progress, as the baseline for comparison. Since it is not clear which factor(s) are represented by “latitude”, we prefer not to present “latitude” as an alternative explanation. Using the same null model approach, we also did examine if day length better explains the observed tracks compared to stochastic **northward** movement *irrespective of day length* (Fig. 2). Since latitude, day length and temperature largely overlap, we previously refrained from testing temperature as an additional alternative explanation for spring migrations of birds. However, following this comment we tested temperature as well, and obtained similar results. This new analysis was added to the main text (Fig. 2, L185-187), further strengthening our conclusions. In addition, we further clarified our use of null model approach by changing the wording in L53-57.

Another issue is the presence of exaggerated statements. I appreciate the value of having a broad inferential scope, but each study must frame their inferences based on the data at hand. Globally there are around 180 species of water fowl, North

America has around 40 species. Only one North American species was considered in this study. Statements in the manuscript that the data is taxonomically and geographically comprehensive are clear exaggerations. The manuscript should provide an objective assessment of what was actually measured and what can be inferred based on the data. I think it would be valuable to state directly in the manuscript that the sample used in this study represents around 6% of the world's water fowl species (this value should be confirmed) with the majority of the data coming from Europe and East Asia.

Response 9: We agree that this study cannot represent all waterfowl in the world, especially North America. We stated this point in the main text (L17, 214, 369-373).

I appreciate the effort made by the authors to expand the Discussion, but the exaggerated statements I identified in the previous version remain. Looking at the first paragraph of the Discussion, I would not describe the data as “exceptionally comprehensive” and I would argue that the study’s findings do not support the conclusion that: “...tracking the green wave is neither a global phenomenon nor a pervasive driver of spring migration of avian herbivores.” I would suggest taking a more objective look at the data and the results, and providing a more balanced and nuanced assessment of the study’s findings that avoids absolute or extreme statements.

Response 10: Previous GW studies have examined only one to three populations of single species. Since we tested 14 populations of 10 species, our analysis is >4-fold more comprehensive, in terms of number of species or populations, than previous studies. We believe this difference is important and should be emphasized, yet a 4-fold increase might not be considered “exceptionally” comprehensive, and also agree that absolute or extreme statements should be avoided. We therefore tuned down this and similar statements throughout the manuscript (L3,17, 26, 72-73, 79, 85, 214, 219-222, 223-224, 225, 290, 320, 331, 341, 369-373, 389-390). See also Responses 32 and 33.

Reviewer #4 (Remarks to the Author):

I commend the work undertaken by the authors to revise their manuscript. Overall, I find it much improved particularly with respect to clarifications in the Introduction setting up the work, and the vastly expanded scope of the new Discussion. The expanded Supplementary material (flow diagram, simulation matrix) is also helpful in orienting the reader through a relatively complex work. In most cases the authors have worked to address the reviews, but in a few cases I believe a little more work is required.

Response 11: Thank you for the kind words and constructive comments!

The expanded discussion within the spatiotemporal context (L223-273) substantially enhances the storyline but there are components that still require some further development:

Responses 67 & 68. The additional text in the Results (L190-194) refer only to the issue of timing, and doesn't seem quite sufficient as is. I am not sure about the statement "This suggests that non-surfers timed their migration irrespective of the green wave". Doesn't it rather indicate they cannot obtain improved green metrics within the overall timing constraints of their migration schedule?

Response 12: We agree and rephrased this point in L205-206. We also supplied results from the spatial aspect, see Responses 13 and 14.

There are related Results elements that only appear in the Discussion (L234-239). These would more sensibly be located at the end of the Results, where they could be more clearly explained (especially the time differences of Suppl. Table 5) and better set the readers up for the ensuing Discussion.

Response 13: We now added relevant text also in the Results (L154-158, 206-211).

The new text (Results and Discussion) particular to the East Asia migrants is welcome; but, please carefully note that the new results reported in the Fig 2 are not identical to previous, with the IRG results for observed East Asian migrants now being reported as “a-b” in several cases. The only clear separations (observed-simulated) using IRG are therefore for the GWFG and the tundra swan. The results do still hold however across East Asia species for the other metrics (GWI and NVDI; Suppl. Fig. 2). Hence the reporting, which ought to close off the Results section, would more correctly be something along the lines of:

“Our results for East Asia indicate that migrants simulated with stochastic stopover sites apparently follow the green wave better than the observed ones. In this flyway, the single grazer (the greater white-fronted goose) and all the facultative herbivores modelled using stochastic stopovers could apparently gain higher green wave index (GWI) values by stochastic migration (Fig. 2 and Suppl. Fig. 2). A similar result holds for the NVDI metric (excepting the bean goose) but is less clearly seen in the IRG.”

Response 14: This is a welcome correction, thank you! We added the suggested text in the revised manuscript (L206-211).

This might be a good place to link to the new maps provided in Suppl. Fig. 3, which are not referred to yet that I can see. Perhaps zooming these to cover the main area of interest (Western Europe- Eastern Asia) might help tell us what is going on with the regional spatiotemporal dynamics, and assist in the Discussion development. Similarly, regarding the new Suppl. Figure 4, the request was to develop a couple of these time series for selected stopover sites for populations that do/do not support the green wave hypothesis, and therefore may be expected to be different (Eg in East Asia cf Barents Sea or Svalbard). A good example of something like this is in Fig. 3 of Aikens et al (2017); here, individual lines may be for birds within populations? Better spatial (SFig3) and temporal (SFig4) representations ought to

help tie the story altogether. This issue was also specifically highlighted by Reviewer 3 (their point 5 - I would also suggest adding figures to the manuscript that summarize the results of the double-logistic regression analyses for each species. The combination of greenness maps and species specific plots would add critical details to the results, which would enhance the depth and quality of the findings and conclusions.) and remains only partially addressed.

Response 15: Thanks for this valuable suggestion. We addressed this by adding the new Supplementary Fig. 2, which includes the migration and its associated greenscape, as well as the double-logistic curve results, for *all* study populations. We further extended the Discussion based on this figure (L138-140, 254-255, 262-265).

Final point on the regional variation (L250-273). Two species not deeply discussed – the Scandinavian bean goose and Svalbard pink footed goose – now appear to show the exact opposite pattern to what is described for the East Asian migrants. That is, observed migrants follow the green wave better than simulated migrants with stochastic stopovers (spatial simulations, new results using IRG, Fig 2). Referring back to the paper’s framework at L465-468: is this then partial support for the green wave hypothesis, i.e., only spatial selectivity for sites with a green wave? c.f. statements at L165, L188, L283 “no significant association”. And if so, are the results for these species relevant to the Discussion about when/how species have capacity to adapt the timing of their ocean crossings versus their ability to select optimal sites en route?

Response 16: We thank the reviewer for this comment, which clarified potential confusion about our results and their interpretation, and also highlighted two interesting cases that demonstrated further insights provided by our approach.

The green wave hypothesis assumes that migrating animals match the spatiotemporal change in plant growth over large scales (i.e. the green wave), requiring migrating animals to reach specific sites (space) in the right timing (time) where and when foliage quality is high¹⁻³. Therefore, matching the green wave only in space or only in time is not sufficient to surf the green wave. Rather, matching in both space and time

is necessary to surf the green wave. We run three types of simulations (stochastic space, stochastic time, and both) to assess in which aspect (space, time or both) birds follow (or not) the green wave during spring, stressing (now with more emphasis, see L529-530) that the GWH requires matching in both space and time, as explained above.

The reviewer highlighted two cases in particular, the Scandinavian taiga bean goose and the Svalbard pink-footed goose. These are indeed two very interesting cases provide important insights. We therefore added a whole new paragraph in the Discussion to highlight, explain and discuss these results L274-289.

The reviewer's question/suggestion that these two cases represent "partial support" for the GWH led us to further explain this terminology. The three terms – "perfect surfer", "partial surfer" and "non-surfer" – were adopted from the definition by Aikens *et al.*¹ and were used *only* for the results of the Simple Correlation, as detailed in L146-150 of the manuscript. These three terms were not applied to the results of the stochastic simulations since, as explained above, the GWH requires matching in both time and space; hence, the simulations either support the GWH or not, they cannot provide a partial support. To avoid further confusions of this kind, we renamed the 'perfect surfer' class as 'surfer', and the 'partial surfer' class as 'weak surfer'.

Other comments:

L218-224. Awkward transition between paragraphs since in both cases ("while another population" and "a notable deviation") the same population is being discussed (East Asian GWFG); needs streamlining.

Response 17: Done (L241-242).

L271. A good place to refer to approaches that systematically manipulate the timing of migration schedules (forwards or backwards in time) rather than the randomisation approach adopted here, for example the space-time-time matrix of Bischof *et al* (2012) *American Naturalist*

<https://www.journals.uchicago.edu/doi/10.1086/667590>.

Response 18: Bischof et al. (2012) provided a good way to quantify the mismatch between the green wave and migration at individual level, and this is indeed a valuable reference for this statement (now added, L310-311). We note, however, that the space-time-time matrix was created within the period of observed migration and the range of observed relocations; and the diagonal was the observed migration as they stated. Therefore, their space-time-time matrix was not developed to systematically manipulate the migration timing. Although we agree that this method can be potentially extended for this purpose, it requires further development and this is beyond the scope of this study.

L298. The utility of multiple metrics seems rather lost in the current version.

Response 19: Done (L342-347).

L367-8 “fewer than five individuals or up to ten stopover sites” is confusing.

Response 20: Reworded as “fewer than five individuals or fewer than ten stopover sites” (L422).

L392. Name the relevant bird populations missing data.

Response 21: Done (L446-447).

Supplementary Table 2, Examples. Preface “Assume” with something like “Evaluating the simple method using stochastic migrations, assume...” and follow “therefore it is a green wave surfer” with “under CSM”. Similarly, preface “When the chance” with something like “Using the Metric Selection approach, when ...”

Response 22: Done.

Supplementary Table 2. Fill all cells for absolute clarity (NAs if not applicable).

Response 23: We now filled all the empty cells.

Supplementary Figure 1. Add “Examples are shown for the STS scheme”.

Response 24: We now clarified this point (LXXX)(1048-1050).

Response 62. The LMM addition to Figure 2 is welcome, although Panels A and B now need their axes explained in the caption. It would still be good to see the full mixed effects model results in a supplementary figure, with a panel per population, showing the population fit (as per Figure 2A) but also the individual-level random slopes. As per the new variance information in Supplementary Table 5 there is very high variance at the individual bird level in some populations.

Response 25: We have now provided explanations for axes of Fig 2A in the caption. No previous study using SCC incorporated year's and individual's random effect on slope, nor did our previous version. However, as this comment emphasized, incorporating the random effect on slope can provide insights into the individual variation in green-wave tracking. We therefore added this random effect on slope and rerun the SCC modeling. The conclusion remains the same with minor changes in the statistics. We accordingly revised the main text (L158-159,496-503), Table 1, Fig. 2, Supplementary Table 4-6, and added the new Supplementary Fig. 2.

Response 64. I appreciate the authors efforts here, though I am surprised by the cited of movement >20km/h at stopovers, since this seems inconsistent with the 5-30km buffers (L405) set at stopover sites for the extraction of green-wave metrics?

Response 26: In order to determine the reasonable buffer size for the extraction of green-wave metrics, we calculated the mean maximum step length within stopover sites (Supplementary Table 3). In this table we can identify in many cases, the movement can be >20km/h within stopover sites, as stated in Response 64 of the initial submission. According to the range of this mean maximum step length within stopover sites, we determined the range of buffer size (5-30km) for sensitivity analyses and for presentation in the manuscript. The values of the buffer size are the radius values. Since the results using different buffer sizes are consistent, we maintained the finest buffer size in the manuscript. We now clarified these points in L460-461).

Response 66. Literally did you try: fit <- lme(...as described); E <- resid(fit, type="normalized")? Also a log transform on the metrics prior could help? If the authors retain their multiple tests approach in favour of an lme approach, somewhere in the manuscript there needs to be an explicit caveat about the unaccounted for autocorrelation (as per my original comment @L397-402).

Response 27: Thank you for suggesting the code! We tried log and power transformation but failed to achieve residual normality unfortunately. We now added the suggested notes in L630-632).

A couple of remaining queries on the dataset:

L341: three sources? Only two stated.

Response 28: Corrected (L395).

L342 “movement information first presented in this study” – and Supplementary Table 3 – are any of the GPS tracks from Eastern China–Eastern Russia birds (GWFG, Swan goose, Bean goose) previously published in the Yu et al (2017) Current Biology paper?

Response 29: Some bird tracks from Eastern China-Eastern Russia were the same birds as Yu *et al* (2017). However, Yu *et al* (2017) only provided within-winter movements and do not refer to spring migration or provide data for this.

Responses 65 and 94: the manuscript text makes reference to exclusion of tracks based on various criteria (eg L366-71; L498). As Supplementary Table 3 currently reads, it looks like this reports the “raw” tracking data, prior to this processing. If it in fact represents all the tracking data to which the SCC is applied, and from which simulations are generated, this could be stated explicitly in the table heading. For example with a sentence like: “Data from a total of 193 birds (222 migration tracks) comprising a total of xxxx GPS locations were retained for use in the correlation analysis and to generate stochastic simulations (n = 1000 each scheme,

see Supplementary xxx)".

Response 30: Supplementary Table 3 reported raw tracks used in SCC and simulation. We added the explanation in the heading of Supplementary Table 3.

Data availability: a statement confirming that all relevant data are available from the authors is the minimum data availability requirement for this journal. "Upon reasonable request" makes it sound like the authors may have some fuzzy (unstated) restrictions? If so these need to be clearly stated.

Response 31: We wrote the Data Availability section in the revised manuscript according to the guidance of the journal and actually one example statement text (<https://www.nature.com/authors/policies/data/data-availability-statements-data-citations.pdf>). However, in September 2018, during our revision of the manuscript, the journal set a higher standard of data reporting (<https://www.nature.com/articles/s41467-018-06012-8>), which now requires the source data file available to reviewers which "should, as a minimum, contain the raw data underlying all reported averages in graphs and charts, and uncropped versions of any gels or blots presented in the figures". The editor also required this file for the next revision. We have fulfilled this new requirement and will also make this source data file publicly available (stated in *Data availability*). This also conforms with the policy of the journal. Beyond this minimum requirement, most raw tracking data are available on Movebank as cited in Supplementary Table 3. Moreover, the coauthors are making the plan to make the raw tracking data in East Asia available in near future.

On a very general note, I suggest choosing a more specific terminology than "avian herbivores" in the title and throughout. Given that the study species are ducks, geese and swans, perhaps migratory waterfowl? Or herbivorous waterbirds?

Response 32: Done (L3, 26, 73, 79, 85, 219-222, 225, 228, 290).

Similarly, the tone could be moderated throughout as some new sections have a

tendency to over-emphasise the point. A few examples here, but the authors should screen the text: L11 “failed” could be “struggle”. L17 and L197 “exceptionally comprehensive” – although 193 birds is admirable, tracking studies are never going to manage to be “comprehensive” relative to population sizes (unless possibly in very tragically endangered cases). Moderate to “large” or “substantial”? Change L297 “robust” to read “conservative”; L290 “superior” to read “reliable” etc.

Response 33: Done (L11, 17, 214, 320, 331, 341). See also Responses 10 and 32.

REFERENCES

- 1 Aikens, E. O. et al. The greenscape shapes surfing of resource waves in a large migratory herbivore. *Ecol Lett* 20, 741-750, doi:10.1111/ele.12772 (2017).
- 2 Shariati-Najafabadi, M. et al. Satellite- versus temperature-derived green wave indices for predicting the timing of spring migration of avian herbivores. *Ecol Indic* 58, 322-331, doi:10.1016/j.ecolind.2015.06.005 (2015).
- 3 Shariati-Najafabadi, M. et al. Migratory herbivorous waterfowl track satellite-derived green wave index. *Plos One* 9, doi:10.1371/journal.pone.0108331 (2014).
- 4 Marjakangas, A. et al. International single species action plan for the conservation of the Taiga Bean Goose *Anser fabalis fabalis*. (Bonn, Germany, 2016).
- 5 Jia, Q. et al. Population estimates and geographical distributions of swans and geese in East Asia based on counts during the non-breeding season. *Bird Conserv Int* 26, 397-417 (2016).

Reviewer #1 (Remarks to the Author):

I have now read the rebuttal letter and revised manuscript entitled "Testing the ubiquity of the green wave as the driver of spring migration of herbivorous waterfowl." I believe the authors have done an adequate job with the revision. I have no further issues with the manuscript except for the same issues (see below) that I and the other reviewers have raised over the course of review.

Issue 1. I still don't like the fact that the authors conclude that green-wave surfing is not ubiquitous, but then conclude that human development likely constraints the ability of these birds to surf the green wave. It just makes the main points of the manuscript blurry, and difficult to interpret. Note though that I thought the evidence and logic behind the human development hypothesis explained in the discussion is much stronger now than before.

Issue 2. I still don't like that the authors don't use a multiple competing hypotheses approach to test the green wave hypothesis. The authors mainly examine the GWH by comparing it to a null model. Reviewer #3 did a good job illustrating why this is an issue. The authors have added a small analysis and a bit of wording in the manuscript, but the framework is still the same. They still state "Rather than testing a range of alternative explanations, we focused on the green wave hypothesis, aiming to provide an unequivocal test of the ubiquity of the green wave as a main driver of avian herbivore migration."

Issue 3. The manuscript really contains a lot of jargon, but I realize that there is little the authors can do about it. Note though that I think the brief explanation of the analyses is very good at the end of the introduction.

Issue 4. The fact that the real relationship between migration and green wave is unknown, it is difficult to assess what method or metric is best. This is inherently difficult in a study like this, and I think the authors do about as good of a job as possible to deal with it. Nonetheless, it makes the manuscript tricky to interpret and kind of muddles the main points of the manuscript.

Reviewer #4 (Remarks to the Author):

I commend the authors for undertaking their second round of revisions diligently. Overall, I find the entire manuscript much improved and the readability much enhanced. In particular, I am very happy with the insertion of the two new Supplementary Figures (2&3) which make the population-level data and the interpretation of the storyline much more accessible to the reader. The ensuing Discussion has a much greater depth, including more coherency at the large scale as well as more

attention to the details of processes likely to be operating in particular regions or affecting particular populations. I also applaud the new Data deposition requirements of the journal, which can enable this valuable data set to be further explored as ecological concepts regarding migratory species evolve in future. This has surely been a large undertaking for the author team to coordinate. There remain a few typos in the text, which I will leave to the editorial process, but otherwise I look forward to seeing the revised paper published. Best regards, Sophie Bestley.

RESPONSE TO REFEREES

Reviewer #1 (Remarks to the Author):

I have now read the rebuttal letter and revised manuscript entitled “Testing the ubiquity of the green wave as the driver of spring migration of herbivorous waterfowl.” I believe the authors have done an adequate job with the revision. I have no further issues with the manuscript except for the same issues (see below) that I and the other reviewers have raised over the course of review.

Response 1: Thanks for your constructive comments throughout the peer-review process. Please see our replies below.

Issue 1. I still don't like the fact that the authors conclude that green-wave surfing is not ubiquitous, but then conclude that human development likely constraints the ability of these birds to surf the green wave. It just makes the main points of the manuscript blurry, and difficult to interpret. Note though that I thought the evidence and logic behind the human development hypothesis explained in the discussion is much stronger now than before.

Response 2: We followed the title suggestion of the editor, slightly modified to “Stochastic simulations reveal few green wave surfing populations among spring migrating herbivorous waterfowl”. This title emphasizes the critical importance of using stochastic simulations in testing the green wave hypothesis, as well as the major conclusion of this study. Indeed, as noted by the reviewer, we suggest that human disturbances constrain the ability of birds to surf the green wave and happy to learn that the reviewer finds this hypothesis logical and better presented, but cannot see in what way this hypothesis contradicts our main conclusion or makes the manuscript blurry and difficult to interpret.

Issue 2. I still don't like that the authors don't use a multiple competing hypotheses approach to test the green wave hypothesis. The authors mainly examine the GWH by comparing it to a null model. Reviewer #3 did a good job illustrating why this is an issue. The authors have added a small analysis and a bit of wording in the manuscript, but the framework is still the same. They still state “Rather than testing a range of alternative explanations, we focused on the green wave hypothesis, aiming to provide an unequivocal test of the ubiquity of the green wave as a main driver of avian herbivore migration.”

Response 3: We remain split regarding this issue, and prefer to provide a strong test to one major hypothesis, rather than a set of tests of multiple alternative explanations which is likely to be weaker due to data limitations. The additions made in the last round suggest that different factors such as human disturbance and air temperature,

and not only green wave indices, should be considered in examining potential drivers of bird spring migration. These insights correspond to the call of the reviewer to examine multiple drivers of migration and not only the green wave, and we further stressed this point in the revised Discussion (L476-479 with 'All Markup' shown in Word).

Issue 3. The manuscript really contains a lot of jargon, but I realize that there is little the authors can do about it. Note thought that I think the brief explanation of the analyses is very good at the end of the introduction.

Response 4: We minimized the use of jargon and hope that the use of a few repeated acronyms will contribute to the readability of this a paper. We thank the reviewer for the kind words re the revised end of the Introduction.

Issue 4. The fact that the real relationship between migration and green wave is unknown, it is difficult to assess what method or metric is best. This is inherently difficult in a study like this, and I think the authors do about as good of a job as possible to deal with it. Nonetheless, it makes the manuscript tricky to interpret and kind of muddles the main points of the manuscript.

Response 5: The reviewer reinforced here the basic challenge of testing the green wave hypothesis, expressed his/her opinion that we addressed this challenge "as good as possible", yet argued that the results are still difficult to interpret. We thank the reviewer for the kind words, but disagree with the last part for five main reasons. First, we covered here more species and populations and larger geographical range than any previous study, to improve representativeness. Second, we introduced the MSSM method which is more theoretically correct than previous methods. Third, we combined these two merits to show that the MSSM method produced the most reasonable results based on basic biology of the studied species and the most basic principles of the green wave hypothesis. Fourth, the MSSM method revealed strong evidence that most herbivorous waterfowl do not track the green wave, only a few grazing populations in specific regions. This result is important both because it calls to carefully consider evidence for the green wave hypothesis, and since it motivates a search for alternative explanations such as the human disturbance hypothesis we raised here. Fifth, our study demonstrates the (much) more powerful practice of testing hypotheses by stochastic simulations rather than by conventional correlations, and this lesson has important implications for other fields as well (e.g. the critical need to control for collinearity, and stochastic simulations as powerful tools to address this problem). However, we agree that large sample size, wider geographical coverage and the use of more advanced tracking devices, could further improve the understanding of the green wave as a driver of bird migration. We further discuss this subject in the Discussion (L460-467 with 'All Markup' shown in Word).

Reviewer #4 (Remarks to the Author):

I commend the authors for undertaking their second round of revisions diligently. Overall, I find the entire manuscript much improved and the readability much enhanced. In particular, I am very happy with the insertion of the two new Supplementary Figures (2&3) which make the population-level data and the interpretation of the storyline much more accessible to the reader. The ensuing Discussion has a much greater depth, including more coherency at the large scale as well as more attention to the details of processes likely to be operating in particular regions or affecting particular populations. I also applaud the new Data deposition requirements of the journal, which can enable this valuable data set to be further explored as ecological concepts regarding migratory species evolve in future. This has surely been a large undertaking for the author team to coordinate. There remain a few typos in the text, which I will leave to the editorial process, but otherwise I look forward to seeing the revised paper published. Best regards, Sophie Bestley.

Response 6: Thank you for your kind words, and for your constructive and insightful comments throughout the peer-review process!